



# 1  Antarctic high-resolution ice flow mapping and increased mass loss

# 2  in Wilkes Land, East Antarctica during 2006–2015

Qiang Shen[1,3], Hansheng Wang[1], Che-Kwan Shum[2,1], Liming Jiang[1,3], Hou Tse Hsu[1],
Jinglong Dong[1,3]
[1]State Key Laboratory of Geodesy and Earth's Dynamics, Institute of Geodesy and Geophysics,
Chinese Academy of Sciences, Wuhan 430077, China
[2]Division of Geodetic Science, School of Earth Sciences, The Ohio State University, Columbus, Ohio
43210, USA
[3]University of Chinese Academy of Sciences, Beijing 100049, China
*Correspondence to*: Qiang Shen (cl980606@asch.whigg.ac.cn)
**Abstract.** Substantial accelerated mass loss, extensive dynamic thinning on the periphery, and
grounding line retreat in the Amundsen Sea Embayment, have amplified the long-standing concerns on
the instability of the Antarctic ice sheet. However, the evolution of the ice sheet and the underlying
causes of the changes remain poorly understood due in part to incomplete observations. Here, we
constructed the ice flow maps for the years 2014 and 2015 at high resolution (100 m), inferred from
Landsat 8 images using feature tracking method. These maps were assembled from 10,690 scenes of
displacement vectors inferred from more than 10,000 optical images acquired from December 2013 to
March 2016. We also estimated the mass discharges of the Antarctic ice sheet in 2006, 2014, and 2015
using the high-resolution ice flow maps, InSAR-derived ice flow map, and the ice thickness data. An
increased mass discharge ($40\pm24$ Gt yr$^{-1}$) from East Indian Ocean sector was found in the last decade,
attributed to unexpected widespread accelerating glaciers in Wilkes Land, East Antarctica, while the
other five oceanic sectors did not show any significant changes, contrary to the long-standing belief
that present-day accelerated mass loss primarily originates from West Antarctica and Antarctic
Peninsula. In addition, we compared the ice sheet mass discharge with the new surface mass balance
(SMB) data to estimate the Antarctic mass balance. The most significant change of mass balance also
occurred in East Indian Ocean during the last decade, reaching $-40\pm50$ Gt yr$^{-1}$, the large uncertainty is
caused mainly by error in the SMB data. The newly discovered significant accelerated mass loss and
speedup of ice shelves in Wilkes Land suggest the potential risk of abrupt and irreversible
destabilization, where the marine ice sheets on an inland-sloping bedrock, are adversely impacted by
increasingly warmer temperature and warm ocean current intrusion, all of which may pose an
unexpected threat of increased sea level rise.

## 37  1  Introduction

A large challenge for rigorous sea-level projection in the 21$^{st}$ century is that the dynamics of the
Antarctic ice-sheet is not sufficiently understood under rapidly warming atmosphere and ocean (Church
et al., 2013; Hanna et al., 2013; Joughin et al., 2012; Pritchard et al., 2012). Recent studies on Antarctic
ice-sheet processes since the 1990s using satellite, airborne and *in situ* observations (McMillan et al.,
2014; Rignot and Thomas, 2002; Shepherd et al., 2012; Vaughan et al., 2013), reported increasing




present-day ice-sheet changes, such as extensive dynamic thinning on the periphery (Pritchard et al., 2009), accelerated mass loss (McMillan et al., 2014; Shepherd et al., 2012), and grounding line retreat in the Amundsen Sea sector, West Antarctica (Rignot et al., 2014), all of which raised the long-standing concerns on ice-sheet instability (Joughin et al., 2014; Vaughan et al., 2013). Although the new observations have greatly improved our ability to quantify the changes to the Antarctic ice sheet, it remains unclear whether East Antarctic ice sheet is losing or gaining mass, especially in the large marine ice sheets of East Antarctica (Mengel and Levermann, 2014). It is also unclear whether the rate of Antarctic ice loss/gain has increased/decreased over the last two decades (Stocker et al., 2014). Furthermore, the underlying drivers of ice-sheet changes remain poorly understood (Alley et al., 2005). All these limitations make it difficult to determine the future behaviour of the ice sheet. The key to understanding the Antarctic ice-sheet dynamics is to more accurately determine its mass budget using extended observations to provide a longer and higher resolution observational record towards improved understanding of the ice-sheet evolutions, which is crucial for more reliable sea-level projections (Hanna et al., 2013; Vaughan et al., 2013).

Glacier ice flow or velocity, one of the critical ice dynamic parameters affecting the estimates of ice sheet mass balance and the corresponding sea level rise (Scheuchl et al., 2012), has been measured by traditional grounded-based measurements (e.g. GPS, electronic distance) since 1970s in the Antarctic ice sheet. However, the sporadic and discontinuous observations prohibit the studying of ice sheet mass balance as a whole. It was not until recently that the glaciologists began to present a complete picture of ice velocity in Antarctica by the use of multi-satellite interferometric synthetic aperture radar (InSAR) with a long data span (1996–2009) (Rignot et al., 2011). However, such a snapshot of ice motion of entire Antarctica is insufficient to provide a clear insight of the spatial and temporal characteristics of ice dynamics. Furthermore, the lack of higher-resolution ice velocity data limits a thorough investigation on localized ice dynamics (Favier and Pattyn, 2015; Nath and Vaughan, 2003), such as crevasse production, role of ice rises on the stability of ice sheet, etc. These limitations highlight the need for a new set of ice velocity observations over Antarctica.

Therefore, here we intend to construct two present-day ice flow maps covering the years of 2014 and 2015 for all of the Antarctica inferred from Landsat 8 (L8) images acquired by the Operational Land Imager (OLI). The velocity data and the existing InSAR-derived ice velocity (Rignot et al., 2011) can be used to estimate the mass discharges in 2006, 2014, 2015 in combination with the Bedmap-2 ice thickness data (Fretwell et al., 2013) associated with IPR (Ice Penetrating Radar) track measurements from the IceBridge project (Allen, 2013, 2011; Blankenship et al., 2011, 2012). Furthermore, the mass balances of the Antarctic ice sheet can be estimated by comparing the mass discharges with the latest ice-sheet SMB data derived from a regional atmospheric climate model (RACMO2.3) (van Wessem et al., 2014) , employing the input-output method (IOM) (Rignot et al., 2008), and the decadal changes can be easily found.

## 2   Data and methods

### 2.1   Data

We collected L8 orthorectified panchromatic bands in 15 m spatial resolution from December 2013 to March 2016 to infer present-day ice velocities of Antarctic ice sheet. The images were acquired by the Operational Land Imager (OLI) on L8 and are managed by the U.S. Geological Survey (USGS) Earth Resources Observation and Science (EROS) Center. L8 is the eighth satellite in the Landsat





missions, launched on February 11, 2013, which provides a continuous series of land and ice surface
observations with 16-day revisit cycle. The OLI has improved radiometric performance in 12-bit
quantization, which can distinguish subtle contrast variations over bright targets (Fahnestock et al.,
2016; Morfitt et al., 2015), such as Antarctica covered only by snow or ice with high reflectivity.
Rigorous calibration and orbital control contribute to the resulting high-quality visible and infrared
images. The OLI is calibrated to <5% uncertainty in absolute spectral radiance and ~8 m geodetic
accuracies (circular error at 90% confidence (CE 90)) (Zanter, 2016).
Compared to the satellite interferometric SAR data, the L8 panchromatic imagery is more suitable to
estimate ice motion in fast-flowing regions for several reasons, (1) the nadir look results in similar
viewing geometry between acquisitions. it can minimize the topographic artifacts, one of main error
sources in SAR/InSAR processing (Mouginot et al., 2012); (2) despite a non-cloud free sensor as
opposed to SAR, L8's 16-day revisit cycle and relative large swaths (185-kilometer), make it possible
to obtain continuous snapshots of ice flow over entire Antarctica, (3) the optical imageries are almost
free of atmospheric effect including ionosphere and troposheric delays, which may introduce errors in
the interferometric SAR imageries, for example, the ionosphere can produce large ice velocity error up
to 17 m yr$^{-1}$ for L-band SAR imagery, (4) the feature tracking method can produce two-dimensional
displacements with same accuracy, while SAR speckle-tracking method has lower accuracy in the
azimuth direction, and differential radar interferometry method only measures one-dimensional
line-of-sight (LOS) displacement.
The level 1 Terrain corrected (L1GT) products packaged as Geographic tagged image file formation
(GeoTIFF) in 16-bit grayscale are used to produce the ice velocity of Antarctica. The L1GT products in
Antarctica are terrain orthorectified data using Radarsat Antarctic Mapping Project Digital Elevation
Model Version 2 (RAMP V2 DEM). The geometrically corrected products have minimal distortions
related to the sensor (e.g., view angle effects), satellite (e.g., attitude deviations from nominal), and
Earth (e.g., rotation, curvature, relief). Radiometric corrections were applied to remove relative
detector differences, dark current bias, and some other artifacts. A complete L1GT product consists of
13 files, i.e., the 11 band images, a product specific metadata file, and a Quality Assessment (QA)
image. In our study, only the panchromatic band, specific metadata file and QA band are used. The
specific metadata are used to obtain the cloud ratio as criteria (40%) to pick images for ice velocity
extraction. The QA band is used to identify the spatial distributions of cloud and water, which are
masked in velocity scenes. Based on visual interpretation and cloud cover ratio, a total of more than
10,000 scenes were selected for producing ice velocities over Antarctica. The projection is polar
stereographic with a true latitude of –71°. The reference ellipsoid used is the WGS84 model. In
addition, for comparison of ice flux and mass balance at different periods, InSAR-derived ice velocity
data (450 m resolution) inferred from multiple satellite InSAR data sets are used. The majority of
InSAR data used are during 2007–2009, but with the data in the grounding lines acquired mostly in
2006 (Rignot et al., 2008; Rignot et al., 2011).
In order to assess the accuracy of our ice velocity results, we also collected *in-situ* measurements





(Brecher, 1982; Frezzotti et al., 1998; Manson et al., 2000; Naruse, 1979; Rott et al., 1998; Skvarca et al., 1999; Zhang et al., 2008), compiled and managed by the National Snow & Ice Data Center (NSIDC). The *in-situ* measurements of ice velocity were obtained from a variety of methods such as differential GPS, electronic distance measurement and triangulation chain survey. The *in-situ* data in the Lambert-Amery basin were obtained mainly from 1988 to 2008, Siple Coast from 1984 to 1998, and Mizuho Plateau of Queen Maud Land from 1969 to 1978. Note that we only collected *in-situ* measurements in the slow-flow regions where ice velocities are less than 100 m yr$^{-1}$ and thus assumed to have no significant secular changes.

## 2.2 Feature tracking method

To determine the horizontal displacement field of ice motion, we use feature tracking method (Bindschadler and Scambos, 1991; Leprince et al., 2007; Scambos et al., 1992), also called as the phase shift method. Since the input images are orthorectified, correlation can be directly implemented using the phase shift technique of low frequency calculated by Fourier-based frequency correlator (Leprince et al., 2007), which is produced within a specific sliding window (or patch) on the paired images repetitively. The result is given by a three-band file consisting the E/W displacement map (positive toward the East), the N/S displacement map (positive toward the North), and the SNR band as an indicator of the quality of the measurement. The technique enables us to resolve sub-pixel displacements of less than 1/20 of the pixel resolution at a high signal-to-noise ratio (SNR), which is generally greater than 0.9. All processes are performed using the COSI-Corr (Co-registration of Optically Sensed Images and Correlation) software package developed at the California Institute of Technology (Leprince et al., 2007).

The feature tracking is implemented in a two-step process. The first step (namely coarse correlation) is to roughly estimate the pixelwise displacement between two patches. In general, if noisy images or large displacements are expected, a larger initial sliding window should be used. In this study, the size of initial sliding window varies from 64 to 256 in pixels in both X and Y directions according to the prior knowledge of InSAR-derived Antarctic ice velocity, and the time interval between two paired images. Once the initial displacements are estimated, the final correlation (namely fine correlation) step is to retrieve the subpixel displacement by using smaller window. The new size of 32×32 pixels is tentatively adopted in order to yield reliable estimates for the displacement at densely independent points. Other parameters of frequency correlator include the step sizes between sliding windows in both X and Y directions (in pixels), frequency masking threshold, the number of iterations for robustness, resampling and gridded output. The step size is set to be a constant value of 7 pixels in each dimension, which means that output product has 100-meter resolution. The frequency masking threshold of 0.9 is adopted as an optimum value as recommended in a previous study (Leprince et al., 2007).

## 2.3 Quality control for displacement vector

During the co-registration step, the Fourier frequency correlator is used in correlation estimates. The technique is more accurate compared with a statistical method; however, it is more sensitive to noise. High-performance L8 images can minimize the effect, but decorrelation still exists due to large ground motion, lack of measureable ground features (such as crevasses, or rise), sensor noise, and topographic artifact (thereby producing imprecise orthorectified data).



To overcome these problems, we devise three steps to enhance the signal and exclude unreliable
measurements. First, we suppress the noise on each displacement scene by using an adaptive filter and
a median filter respectively. The adaptive filter is the local sigma filter (Eliason and Mcewen, 1990).
The filter size is 9 pixels and sigma factor is 2. A median filter is further applied to remove "salt and
pepper" noise in ice displacement scene. Second, the areas covered by cloud and water are excluded
from the displacement scenes using the QA band (Zanter, 2016). In the QA band, each pixel contains
16-bit integer that represents bit-packed combinations of surface, atmosphere, and sensor conditions at
different confidence level. The pixels covered by cloud and water in paired images are unpacked from
the QA band using our developed procedures, the pixels marked by cloud and water at high confidence
(67–100%) are used to build a mask layer, and then they are masked from displacement scenes. It
should be noted that the identification of cirrus is problematic in raw images based on our analysis;
snow and ice are easily considered as cirrus. Here, we only use cloud to build mask layer. Third, since
the frequency correlation easily gives rise to errors at the edges of displacement image, the pixels are
also masked.

### 2.4 Ice velocity extraction

The cloud contamination is a main challenge in ice flow generation using optical images. In order to
overcome the problem, we process all image pairs using a one-year time intervals as time baseline with
the minimum repeat cycle of 16 days in Worldwide Reference System (WRS-2). Some adjacent paths
in WRS-2 are also paired to produce ice velocity for some void areas where there are no valid scenes
for pairing in same path and row. The one-year time interval is derived from our correlation
experiments. When the time interval is more than one year, the decorrelation may appear due to large
surface motion or geomorphic change. Finally, 10,690 image pairs are selected from more than 10,000
scenes of L8 panchromatic images, and processed for the production of ice velocity.
Despite of the improvement in geometric accuracy, the residual geolocation errors with L8
panchromatic band still exists (~8m) in CE90, the errors could cause offsets between the displacement
scenes which should be removed (Fahnestock et al., 2016). In fact, the offset tuning is often called
absolute calibration of the ice velocity data. In Antarctica, absolute calibration is a challenging issue
because the ice is active almost in everywhere and available rock outcrops are extremely scarce. Here,
we use the InSAR-derived Antarctic velocity map to determine the relatively stagnant areas (the
magnitude of ice velocity <5 m yr$^{-1}$) for absolute calibration of our ice velocity estimates.
There are three steps for the velocity calibration. First, the differences of the displacements between
InSAR-derived velocity map and our displacements are calculated in the stagnant areas. Second, to
eliminate outliers, a three-sigma filter is applied recursively on these differences in which the
differences will be omitted if the magnitudes of the values are larger than three times the standard
deviation (3$\sigma$). Third, the mean of the rest differences is considered as the offset of displacement
scenes. Furthermore, the offsets for the displacement scenes outside of stagnant areas (such as in the
Ross and Ronne ice shelves) are estimated by overlapped neighboring scenes at nearly the same time
periods. The two velocity components are independently estimated and rectified.
The mosaicked velocity map is produced on the basis of processed displacement scenes as above. To





increase the accuracy of mosaicked velocity map, we stack all displacement scenes after the pixels with
SNR < 0.9 are masked. In general, the velocity map contains 8–10 scenes in a specific location. For a
specific pixel denoted by $i$, all displacement scenes ($m$=1, 2, …, $n$) are stacked to give the estimate of
the ice velocity ($V_i$) as follows,
$$V_i = \frac{\sum_{m=1}^{n} \Delta d_m^i}{\sum_{m=1}^{n} \Delta t_m^i}$$

208                                                                                                     (1)

where $\Delta d_m^i$ denotes the generated displacement during the time interval $\Delta t_m^i$.

### 3   Results of ice velocity

### 3.1 Ice velocity maps

In Antarctica, the valuable L8 images are available just in summer (November, December, January,
February and March). Due to the short observational span at the end of 2013 and at the beginning of
2016, it is difficult to produce the individual mosaic for the entire Antarctica, thus the images acquired
in the two years are used to produce 2014 mosaic and 2015 mosaic respectively. In Figure 1, we show
two mosaicked ice velocity maps for 2014 (Fig. 1a) and 2015 (Fig. 1b), respectively for Antarctica. Ice
velocity differences between the two maps are usually very small relative to the magnitudes of the
velocities since the mean and standard deviation are 0.17 m yr$^{-1}$ and 7.6 m yr$^{-1}$(Fig. 1d). The
InSAR-derived ice velocity data are also shown (Fig. 1c), in which the data at grounding lines used for
the analysis of glacier discharge changes were derived from the SAR images in about 2006 (Rignot et
al., 2011). Our results exhibit a similar pattern in the ice flow field compared with a previous
InSAR-based study over a long time span (Rignot et al., 2011). The spatial resolution of our velocity
maps is 100-m, which is 4 times higher than the InSAR-derived ice flow map. Our two ice velocity
maps thus provide an opportunity to investigate localized ice dynamics, such as crevasse production,
and the roles of ice rise and rumples on ice-sheet dynamics and evolution. They also have a better
coverage over Antarctica except for the south of 82.5S. The two mosaicked ice velocity maps cover the
majority of the ice sheet and nearly 99% of fast-flow glaciers and ice shelves, and fast ice, except for a
few ice streams of Ronne ice shelf (e.g. Academy, Foundation glaciers) and Ross ice shelf (e.g.
Whillans glacier in Siple Coast). In addition, in order to be computationally efficient, the entire
Antarctica is divided into 11 sub-regions, and data stacking is processed independently, then 11 sub
regions (Fig. 7) are mosaicked to generate an ice velocity map for the entire Antarctica.





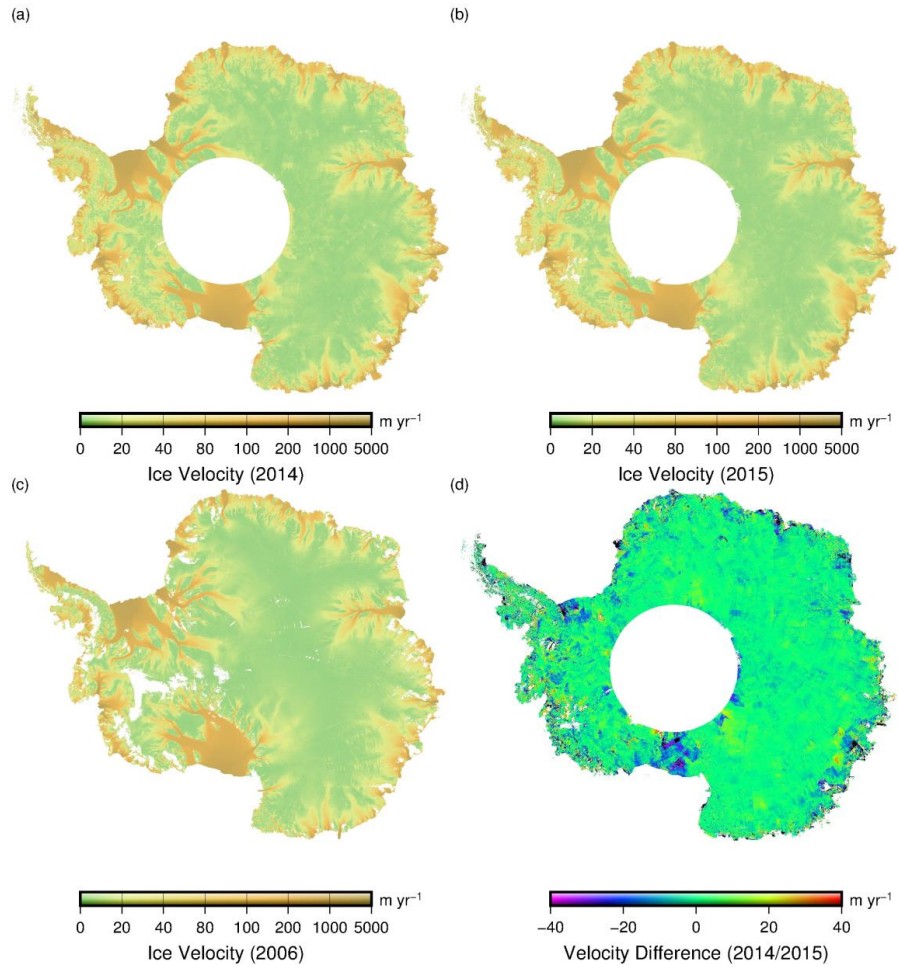

**Figure 1.** L8-derived '2014' ice flow in a) from December, 2013 to December, 2014, and L8-derived '2015' ice flow in b) from January, 2015 to March, 2016, and InSAR-derived ice velocity in c) from 1996 to 2009; the difference of ice flow between '2015' and '2014' in d). L8-derived ice velocity maps are drawn with a resolution of 500m.

### 3.2 Uncertainty analysis

The uncertainty of ice velocity maps derived from the L8 data primarily resulted from mis-registration, the time interval of the pairs used to extract displacement, and the amount of stacking data. The mis-registration is mainly caused by three error sources: (1) decorrelation due to severe ground change, lack of measureable features between the scenes due to long time interval or single land cover (e.g. snow or ice); (2) low image quality caused by sensor noise, pixel oversaturation, aliasing and cloud contamination; (3) topographic artifacts primarily due to shadowing differences and imprecise orthorectification of satellite attitudes. The co-registration accuracy is conservatively set to be 1/25 of the pixel size in E/W and N/S displacement components, which is larger than 1/50 of the pixel size





prompted by Leprince et al. (2007) (Leprince et al., 2007). Using the co-registration error together with
the total amount of stacking data, and time interval between two acquisitions, the ice velocity error per
year can be calculated on the basis of error propagation law.
According to the mosaicking method as mentioned above (Eq. 1), the uncertainty of one mosaicked
velocity component at i-th pixel denoted by $\sigma_{V_i}$ can be estimated using the following error
propagation formula under the assumption that the errors of different sources are independent:

$$\sigma_{V_i} = \pm \sqrt{\frac{\sum_{m=1}^{n} (\sigma_m^i)^2}{\left(\sum_{m=1}^{n} \Delta t_m^i\right)^2}}$$

253    (2)

Where $\sigma_m^i$ is the co-registration error, i.e., the standard deviation of m-th displacement observation
during time interval of $\Delta t_m^i$. Since the co-registration errors are constant in space (the whole scene)
and time domain (all stacked displacements), if the $\sigma_m^i$ is denoted by a constant of $\sigma$, Eq. 2 can be
simplified as follows,

$$\sigma_{V_i} = \pm \sqrt{n} \frac{\sigma}{\sum_{m=1}^{n} \Delta t_m^i}$$

258    (3)

The uncertainty of a mosaicked velocity map is dependent on the amount of stacking data and the time
intervals during the velocity stacking. That means that the larger the time span, the higher resulting ice
velocity accuracy. Since the E/W and N/S components at i-th pixel have the same uncertainty, the
uncertainty as calculated with Eq. 3 is actually valid for the magnitude of the velocity vector. The error
of magnitude of mosaicked velocity vector with magnitudes of 0-20 m yr$^{-1}$ is shown in Figure 2a. For
comparison, the uncertainty of InSAR ice velocity is also shown in Figure 2b.





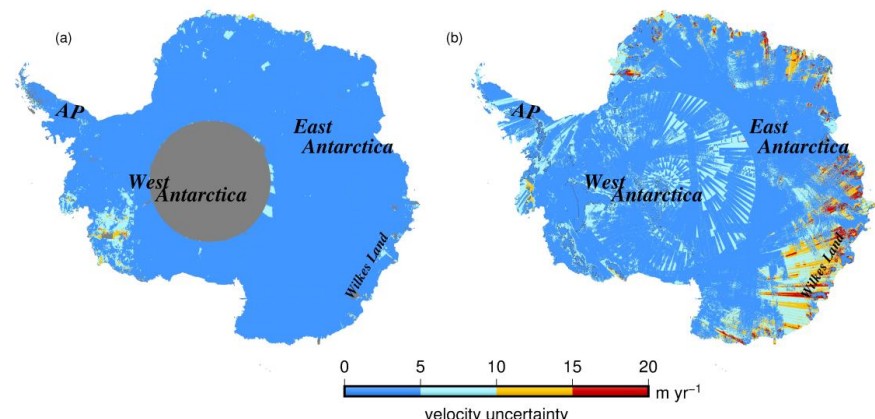

**Figure 2.** The uncertainty maps of L8-derived Antarctic ice velocity in 2015 (a) and InSAR-derived ice velocity (b).

### 3.3 Comparison with *in-situ* measurements

Our ice velocity results are only compared with the *in-situ* measurements in the slow-flow areas (<100 m yr$^{-1}$). The 538 sites chosen for the comparison are shown by the dots of Figure 3, and the differences are shown by the color dots. From the upper inset, the differences are usually <10 m yr$^{-1}$ and the average of the difference is 3 m yr$^{-1}$ with a standard deviation of 10 m yr$^{-1}$. For comparison, the differences between InSAR velocity and field surveying data are also shown in lower inset in Figure 3. The average of the difference is 0.3 m yr$^{-1}$ with a standard deviation of 4.2 m yr$^{-1}$. The differences in accuracy performance may result from the measurement errors and different time spans of surveys.





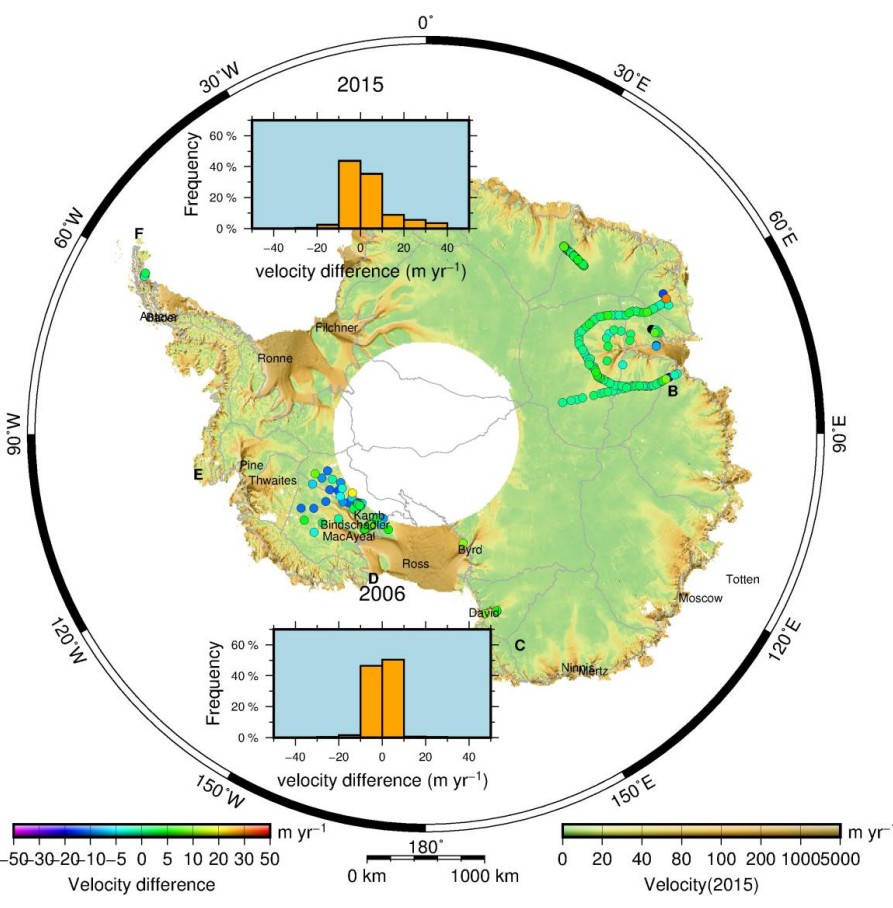

**Figure 3.** The comparison between L8-derived ice velocities in 2015 and data from *in-situ* measurements. The colored dots show the differences between L8 ice velocity in 2015 and *in-situ* survey data. Upper inset shows the histogram of differences between L8-derived ice velocity and field data, and lower inset also shows the same but for InSAR-derived ice velocity.

## 4 Decadal glacier dynamics

We investigated the decadal evolution of ice dynamics of 465 glaciers, nearly all of the glaciers in Antarctica, based on our estimate of high-resolution ice velocity maps in 2015 and an InSAR-derived map in 2006 (Rignot et al., 2011) (Table S1, supplementary materials). 218 glaciers were found to be accelerating, and only 82 glaciers underwent decelerations at a high confidence level ($2\sigma$) (Fig. 4). We found significant outlet glacier accelerations (>50% in velocity change, same hereafter) over much of the Antarctic Peninsula (AP), nearly 30% in Ellsworth Land in West Antarctica, and approximately 25% in the Victoria Land and the Wilkes Land in East Antarctica. In contrast, glacier decelerations were found at rates of 20–40% in the Dronning Maud Land, and at 3–20% for the glaciers in the three largest ice-shelf systems (Filchner-Ronne, Ross, and Amery). In particular, majority of the glaciers accelerated by more than 200% in the northern part of the Western AP (WAP) along the Bellingshausen Sea coast, resulting from the intrusion of the warmer Circumpolar Deep Water (CDW) (Martinson et al.,





2008; Smith et al., 1999) and increasing air temperature (Vaughan et al., 2003). The acceleration in
WAP is more significant than that over the period 1993-2003 (Pritchard and Vaughan, 2007). In the
Weddell Sea sector, velocities of most of the glaciers in the Eastern AP (EAP) and Coats Land in East
Antarctica evidently accelerated by 5–50%, whereas the glaciers draining into the Filchner and Ronne
ice shelves exhibited deceleration. We found complicated decadal variations of glacier dynamics in
East Antarctica. In the West Indian Ocean sector, Dronning Maud Land glacier velocities exhibited an
overall deceleration, whereas its adjacent region, Enderby Land, exhibited acceleration by 5–30%.
However, in the East Indian Ocean sector, most glaciers accelerated by ~25%. In the Ross Sea sector,
the glaciers in Victoria Land accelerated widely by ~20%, whereas much of the glaciers draining into
the Ross ice shelf decelerated, especially for the fast-flowing large Byrd and Mulock glaciers in the
Transantarctic Mountains, Bindschadler and other unnamed glaciers in the Siple Coast. In the
Amundsen Sea sector, although the Pine Island and Thwaites glaciers evidently accelerated by ~15%,
many of the remaining glaciers draining into the Getz ice shelf decelerated. The details of ice dynamics
on individual glaciers can be found in the supplementary materials.

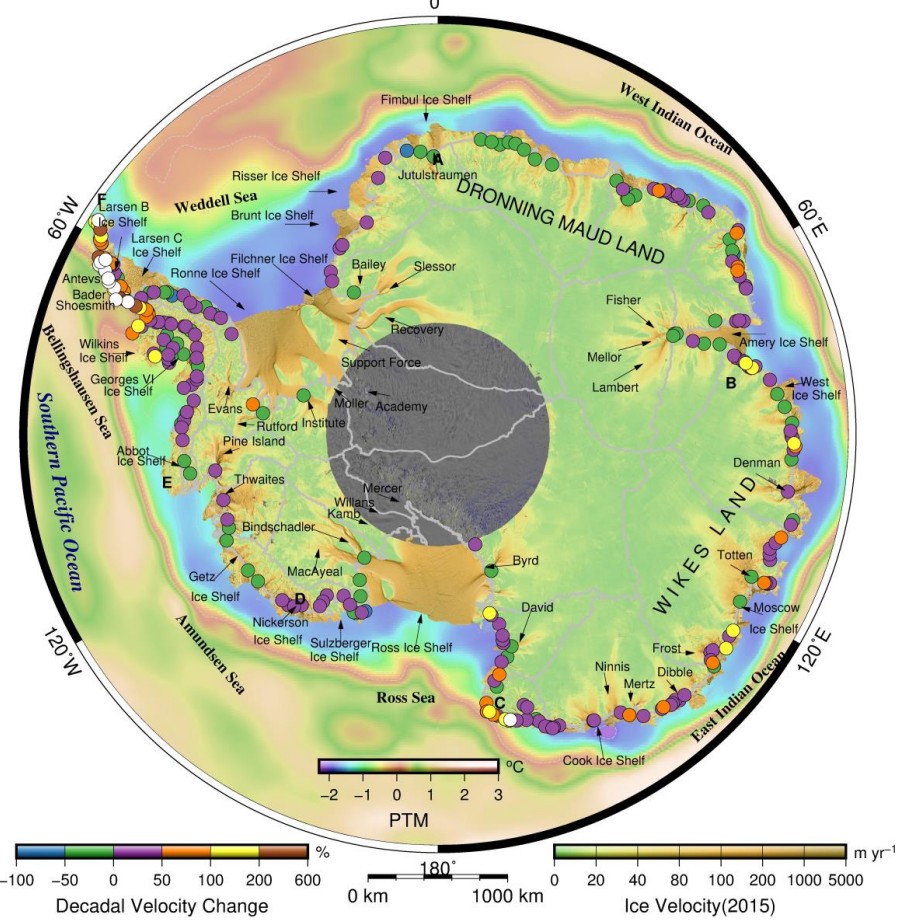


**Figure 4.** Ice velocity in 2015 and decadal velocity change in Antarctica. The mosaic of the Antarctic
ice velocity (2015) derived from L8 panchromatic images from January 2015 to March 2016 is shown





here overlaid on a MODIS mosaic of Antarctica (MOA)(Bohlander and Scambos, 2007). The
magnitude of ice velocity is coloured on a logarithmic scale and overlaid on gridded potential
temperature data of seawater (PTM) at a depth of 200 m from the World Ocean Circulation Experiment
(WOCE). The white-filled dots show that the velocity changes are larger than 600%. The velocity
changes on grounding lines are calculated for 465 glaciers between 2015 and 2006, and shown for the
300 glaciers with high confidence levels (>2$\sigma$) coloured on a logarithmic scale. The names of
selected glaciers and ice shelves are labelled. 'A' through 'F' delimits the six oceanic sectors. The
details of ice velocity changes on grounding lines are presented in Table S1. The solid grey lines
delineate major ice divides.

**5     Decadal variations of mass discharge and mass balance**

We use ice flow measurements for 2014 and 2015 and the existing InSAR results for 2006 to infer
the corresponding Antarctic ice sheet losses at the drainage basin scale (Zwally et al., 2012) in
combination with Bedmap-2 ice thickness data (Fretwell et al., 2013) and ice penetrating radar (IPR)
thickness from multiple campaigns from 2002 to 2014 from the IceBridge project (Allen, 2013, 2011;
Blankenship et al., 2011, 2012) (see supplementary materials). We compare the ice sheet discharge with
the new surface mass balance (SMB) data (1979–2014) (van Wessem et al., 2014) to estimate the
Antarctic mass balance using input-output method (Rignot et al., 2008). The mass discharges across the
Antarctic grounding lines (Depoorter et al., 2013) are derived from the flux gate method (Rignot et al.,
2013) using a developed procedure (see supplementary materials). Here, we calculate the ice-sheet
inflow mass using the new SMB data at a horizontal resolution of 27.5 km resulting from the updated
regional Atmospheric Climate Model RACMO2.3 on the 27 glacier drainage basins (Zwally et al.,
2012) (Table S3). Figure 5 shows the mass discharge, mass balance and their changes between 2015
and 2006 covering the entire Antarctic ice sheet. The total mass balance estimates of the Antarctic ice
sheet under the constant accumulation rate (Monaghan, 2006) during the survey period were –181±68
Gt yr$^{-1}$, –232±60 Gt yr$^{-1}$, and –230±60 Gt yr$^{-1}$ in 2006, 2014 and 2015, respectively (Table 1, S2). These
results are comparable with the latest results inferred from GRACE (Williams et al., 2014) and
Cryosat-2 (McMillan et al., 2014) data, and consistent with recent InSAR mass blance estimates in
2006 (Rignot, 2008). However, our estimated rates are larger than the previous results obtained using
ICESat altimetry data (Shepherd et al., 2012). Table S4 shows detailed estimates of mass balance using
altimetry, gravimetry, and IOM method in the last several decades. The Amundsen Sea sector had the
largest imbalance of –212±24 Gt yr$^{-1}$ in 2015 (similar to previous studies (Vaughan et al., 2013)),
accounting for nearly the total imbalance (–230±60 Gt yr$^{-1}$) of the entire Antarctic ice sheet. Besides
the Amundsen Sea sector, another significant negative imbalance (–78±32 Gt yr$^{-1}$) was observed in the
East Indian Ocean sector of East Antarctica, whereas the West Indian Ocean sector exhibited an
obvious positive mass balance (64±29 Gt yr$^{-1}$). The Ross Sea sector exhibited slight mass gain, whereas
the Weddell and the Bellingshausen Sea sectors exhibited no significant mass changes. However, the
mass balance estimates in the Bellingshausen Sea sectors are most likely underestimated owing to the
summer meltwater not being considered (see supplement materials). Ice-sheet mass budgets are also
shown in Table 2 for East Antarctica, West Antarctica and the Antarctic Peninsula. On the East
Antarctic ice sheet, no significant mass change occurred (11±86 Gt yr$^{-1}$), similar to recent estimates
(Shepherd et al., 2012), because the net mass deficit in the East Indian Ocean sector was compensated
by the mass gain in the West Indian Ocean sector. In West Antarctica, the total mass balance was –
274±41 Gt yr$^{-1}$ in 2015, which is larger than recent altimetry estimates (–134 Gt yr$^{-1}$) from 2010 to
2013 (McMillan et al., 2014) and much larger than recently reconciled estimates (–65±26 Gt yr$^{-1}$) from



2003 to 2009 (Shepherd et al., 2012). In the Antarctic Peninsula, there was a positive mass balance
(33±21 Gt yr$^{-1}$) in 2015, contrary to previously studies (Rignot et al., 2008; Shepherd et al., 2012),
probably due to a larger estimate of snow accumulation rate from the new high-resolution (5.5 km)
SMB data (van Wessem et al., 2016), and the summer meltwater not included(see supplement
materials).

**Table 1.** Mass budgets for the six oceanic sectors of the Antarctic ice sheet

| Oceanic Sector | Area | SMB | GLF(2006) | GLF(2014) | GLF(2015) | Net(2006) | Net(2014) | Net(2015) | GLL |
|---|---|---|---|---|---|---|---|---|---|
| | km$^2$ | Gt yr$^{-1}$ | Gt yr$^{-1}$ | Gt yr$^{-1}$ | Gt yr$^{-1}$ | Gt yr$^{-1}$ | Gt yr$^{-1}$ | Gt yr$^{-1}$ | km |
| Ross Sea (ROS) | 2763447 | 191±12 | 171±11 | 171±6 | 171±6 | 20±16 | 20±13 | 20±13 | 8334 |
| Amundsen Sea (AMU) | 590119 | 319±24 | 524±5 | 532±5 | 531±5 | -205±24 | -213±24 | -212±24 | 4481 |
| Bellingshausen Sea (BEL) | 206768 | 221±11 | 226±4 | 232±3 | 229±4 | -5±11 | -11±11 | -8±11 | 5295 |
| Weddell Sea (WED) | 3240372 | 393±26 | 396±23 | 406±11 | 407±11 | -3±34 | -13±28 | -14±28 | 12138 |
| West Indian Ocean (WIS) | 2544605 | 267±29 | 216±13 | 210±7 | 203±6 | 51±31 | 57±29 | 64±29 | 7978 |
| East Indian Ocean (EIS) | 2549133 | 511±32 | 549±23 | 581±6 | 589±4 | -38±39 | -70±32 | -78±32 | 7213 |
| Total in Antarctica | 11894445 | 1901±58 | 2082±37 | 2133±16 | 2131±16 | -181±68 | -232±60 | -230±60 | 45439 |

The glacier mass discharge or grounding-line flux is denoted by 'GLF', the mass balance by 'Net' is
SMB minus GLF, and grounding line length by 'GLL'. The results for 2014 are given for the period
from December 2013 to December 2014, and 2015 from January 2015 to March 2016. The ice-sheet
area (Area) excludes ice rises and islands, which isolate the main ice sheet. The details about the
glacier's affiliation to the six oceanic sectors can be found in the supplementary materials.

**Table 2**. Ice-sheet mass budgets of East Antarctica, West Antarctica and the Antarctic Peninsula

| Sector | Area | SMB | GLF(2006) | GLF(2014) | GLF(2015) | Net(2006) | Net(2014) | Net(2015) | GLL |
|---|---|---|---|---|---|---|---|---|---|
| | km$^2$ | Gt yr$^{-1}$ | Gt yr$^{-1}$ | Gt yr$^{-1}$ | Gt yr$^{-1}$ | Gt yr$^{-1}$ | Gt yr$^{-1}$ | Gt yr$^{-1}$ | km |
| East Antarctica | 9915811 | 1106±84 | 1066±52 | 1097±21 | 1095±19 | 40±98 | 9±86 | 11±86 | 25697 |
| West Antarctica | 1747718 | 578±39 | 846±21 | 856±13 | 852±15 | -268±44 | -278±41 | -274±41 | 12130 |
| Antarctic Peninsula | 230916 | 217±12 | 170±6 | 179±3 | 184±3 | 47±13 | 38±12 | 33±12 | 7612 |
| Total in Antarctica | 11894445 | 1901±93 | 2082±56 | 2133±25 | 2131±24 | -181±108 | -232±96 | -230±96 | 45439 |


We further analyzed the decadal change of mass balance in the Antarctic ice sheet from 2006 to 2015
(Fig. 5). The mass balance decreased by 27% during the last decade to reach a rate of -230±60 Gt yr$^{-1}$
in 2015, compared with -181±68 in 2006. The change of mass balance (-49±90 Gt yr$^{-1}$) is not
significant in comparison to its large uncertainty caused mainly by SMB. The most significant change
of mass balance occurred in East Indian Ocean, reaching -40±50 Gt yr$^{-1}$. We found an increased mass
discharge from East Indian Ocean sector in the last decade by up to 40±24 Gt yr$^{-1}$, attributed to
unexpected widespread accelerating glaciers in Wilkes Land, East Antarctica. The underlying cause for
this accelerated mass discharge is most likely linked to the incursion of warm CDW towards glacier
termini and a reduction in sea ice (Miles et al., 2016). In Wilkes Land, the large accelerated mass
discharge, together with anomalous glacier retreat (Miles et al., 2016), the contemporary thinning along
its margins (Pritchard et al., 2012) and unstable inland-sloping bedrock topography, suggests potential
instability of the marine ice sheet in a warmer temperature and warm ocean current environments (Gille,
2002; Vaughan et al., 2013). These results are inconsistent with the previously documented persistent
state for the last 14 Myr (Aitken et al., 2016). The Aurora Subglacial Basin (ASB) in the western





Wilkes Land is located to the northeast of elevated Dome A and Ridge B on the Antarctic ice sheet (Fig.
6). The ASB is overlain by 2–4.5 km of ice, which holds an ice mass equivalent to 9 m of sea level rise.
IPR data have identified a series of deep topographic troughs (more than 1 km below sea-level) within
a mountain block landscape oriented nearly orthogonal to the modern margins (Young et al., 2011). The
accelerated mass discharge at the margins of the ice sheet may trigger instability of the upstream ice
sheet (e.g., ASB), which has happened many times throughout the paleo-climate era and has
significantly contributed to sea level changes (Young et al., 2011). In the Wilkes Subglacial Basin
(WSB) of the Eastern Wilkes Land, which holds marine ice equivalent to 19 m of sea-level rise
(Mengel and Levermann, 2014), the marginal glaciers (e.g., Cook, Ninnis) which function as an ice
plug supporting the marine ice sheet of WSB also exhibit obviously accelerated mass discharge. In
contrast, the other five sectors exhibit no significant mass discharge changes. Interestingly, in Pine
Island and the Thwaites catchment (basins 21, 22), West Antarctica, and the Antarctic Peninsula, the
accelerated mass discharges are observed to be $13\pm1$ Gt yr$^{-1}$, $10\pm25$ Gt yr$^{-1}$ and $14\pm7$ Gt yr$^{-1}$,
respectively, in the past 10 years, which are obviously less than the previous estimates of $46\pm5$ Gt yr$^{-1}$,
$46\pm23$ Gt yr$^{-1}$ and $29\pm13$ Gt yr$^{-1}$ in 1996-2006 (Rignot et al., 2008). However, the underlying causes are
unclear in these regions.

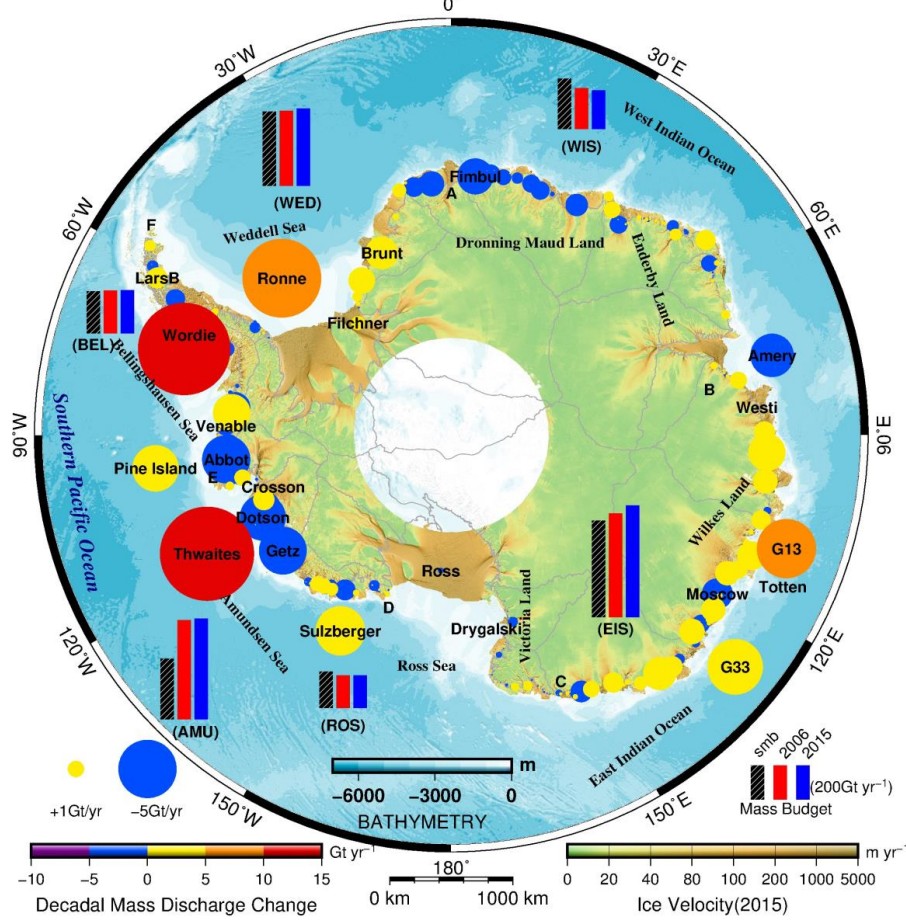




**Figure 5**. Changes of mass discharges and mass balances over Antarctic ice sheet between 2006 and 2015. The colour and size of the circles denote the magnitudes of the decadal mass discharge changes for individual glaciers with no ice-shelf linked and for the combinations of the glaciers linked the same ice shelves. Note that the circles are drawn in variable size scales for clarity. The details about the glaciers can be found in Table S2. In addition, the SMB, the mass discharges in six oceanic sectors in 2015 and 2006 are denoted by black-hatched and coloured bars in six oceanic sectors. The mosaic of ice velocity in 2015 and ice divides, as in Figure 4, and an overlain bathymetry map are shown. The six oceanic sectors include Ross Sea (ROS), Amundsen Sea (AMU), Bellingshausen Sea (BEL), Weddell Sea (WED), West Indian Ocean (WIS) and East Indian Ocean (EIS).

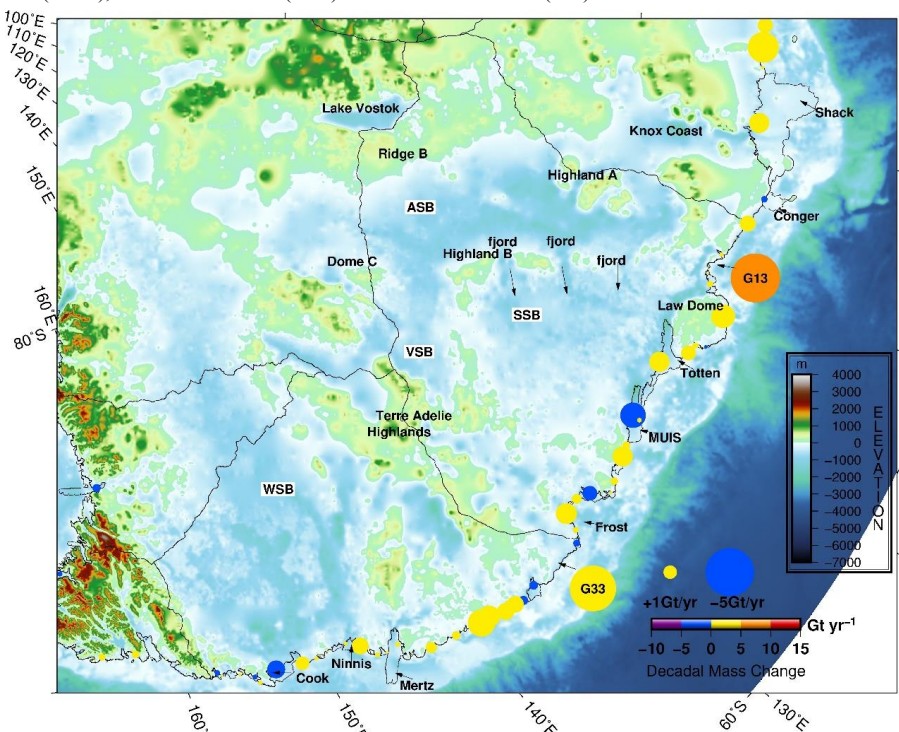

**Figure 6.** Bed topography of the Wilkes Land, East Antarctica. Color circles show the mass balance changes between 2015 and 2006 for individual glaciers with no ice-shelf linked and for glacier combinations in view of the same linked ice shelves. ASB: Aurora Subglacial Basin, VSB: Vincennes Subglacial Basin, VST: Vanderford Subglacial Trench, SSB: Sabrina Subglacial Basin, WSB: Wilkes Subglacial Basin.

## 6   Simultaneous acceleration of ice shelves and glaciers

The velocity changes of ice shelves are also investigated to reveal their underlying relationships with linked glaciers, since the Antarctic ice shelves can be seen as the ice plugs of their bounded ice-sheets and tributary ice-streams, effectively deterring their retreat or abrupt disintegration (Mengel and Levermann, 2014). 204 of the surveyed ice shelves (Table S1, Fig. 7) were found to accelerate mainly in Wilkes Land in the East Indian Ocean sector, Enderby Land in the West Indian Ocean sector, and WAP in the Bellingshausen Sea sector. This result suggests that acceleration of ice shelves is a possible




cause of the fast flow of glaciers. Especially in Wilkes Land, the glaciers and corresponding ice shelves
exhibited nearly simultaneous acceleration. This acceleration further enhances the concerns of the
instability of the marine ice sheets in Wilkes Land, East Antarctica. The marine ice sheets in Wilkes
Land hold ice equivalent to more than 28-m of global sea-level rise, which is more than six times that
of West Antarctica (Mengel and Levermann, 2014).

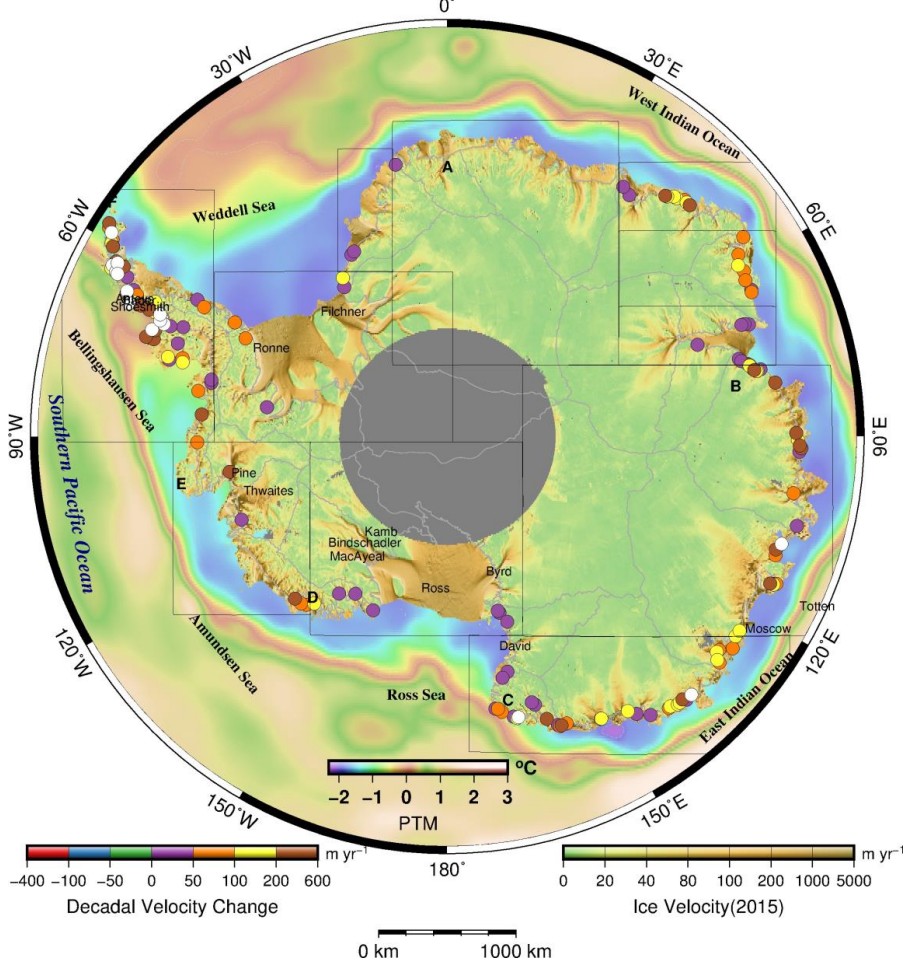


**Figure 7.** The velocity change of the Antarctic ice shelves between 2015 and 2006. The color dots
show the velocity changes of Antarctic ice shelves. The white dots show the changes of ice-shelf
velocity are larger than 600 m yr⁻¹. However, in western Antarctic Peninsula, the ice velocity changes
are shown mostly for glaciers. The mosaic of present ice velocity for 2015 and a gridded potential
temperature data of seawater (PTM) at 200 m depth are also shown as background. The boxes show the
11 mosaicked sub-regions for ice velocity.

**7  Conclusions**
In this contribution, we constructed two high-resolution ice flow maps covering the years of 2014





and 2015 for the entire Antarctica, which can accurately describe the current ice dynamics in the area.
We also found a significantly increased mass discharge in East Indian Ocean sector by 40±24 Gt yr$^{-1}$
over the last decade, attributable to the widespread accelerating glaciers in Wilkes Land, East
Antarctica, while the other five oceanic sectors did not show obvious changes in mass discharge,
contrary to the long-standing belief that present-day accelerated mass loss primarily originates from
West Antarctica and Antarctic Peninsula. In the entire Antarctic ice sheet, total mass balance decreased
by 49±90 Gt yr$^{-1}$ during the last decade, a decline of 27% from 2006 (-181±68 Gt yr$^{-1}$). The most
significant change of mass balance was found in East Indian Ocean during the last decade, reaching
-40±50 Gt yr$^{-1}$. The large uncertainty of mass balance change is mainly due to error in the SMB data.
The significant increased mass discharge together with synchronized speedup of the linked ice shelves
in Wilkes Land suggests a potential risk of destabilization of the marine ice sheet in the region overlain
by the large subglacial basins with inland-sloping bedrock and deep troughs, an instable bedrock
configuration like West Antarctica. Our new high-resolution ice flow maps together with existing
InSAR ice velocity allow the first continent-wide assessment of ice flow and ice discharge changes
during the last decade, which will contribute to our understanding of the entire Antarctic ice dynamics,
and to potentially improving ice-sheet modelling and sea-level projections in the 21$^{st}$ century.
Supplement materials include:
Table S1, S2, S3, and S4 in Excel format
Supplementary materials in pdf format
**Data availability**
The data used in this paper include ice velocity data, ice thickness data, optical satellite images and
grounding line products. The ice velocity data are from
http://nsidc.org/data/docs/measures/nsidc0484_rignot/. http://nsidc.org/data/nsidc-0545/index.html, and
http://nsidc.org/data/velmap/; to obtain our velocity products, please contact the author. The ice
thickness products are provided by the Bedmap program from
http://www.antarctica.ac.uk/bas_research/data/ access/bedmap/download/, and by IceBridge (IPR data)
from http://nsidc.org/data/docs/daac/icebridge/irmcr3/,
https://nsidc.org/data/docs/daac/icebridge/brmcr2/, http://nsidc.org/data/ir1hi2,
https://nsidc.org/data/ir2hi2. The optical satellite images are from http://glovis.usgs.gov and
http://earthexplorer.usgs.gov/ for Landsat. Other data are from
http://icesat4.gsfc.nasa.gov/cryo_data/ant_grn_drainage_systems.php for Antarctic drainage basins,
http://nsidc.org/data/atlas/news/antarctic_coastlines.html for the MOA grounding line and coastline,
http://nsidc.org/data/nsidc-0489 for the ASAID project grounding line,
http://nsidc.org/data/docs/measures/nsidc0498_rignot/ for the DInSAR grounding line,
http://pangaea.de/ for the grounding line provided by Depoorter et al.(2013),
http://nsidc.org/data/nsidc-0082 for the Antarctica DEM, and http://www.staff.science.uu.nl/ for the
FDM and SMB data, http://nsidc.org/data/docs/agdc/nsidc0280/index.html for MODIS Mosaic of
Antarctica(Bohlander and Scambos, 2007), and http://woceatlas.tamu.edu/ for potential temperature of
seawater and bathymetry.
**Author contribution**
Q.S. conceived the study, analysis and wrote the article. H.W. contributed to the research framework,





helped develop the methodology and helped interpret the results and edited the manuscript. C.K.S. contributed to the interpretation of the results and edited the manuscript. L.J. contributed to the data analysis. H.T.H. contributed to the result analysis. J.D. contributed to the data processing. All authors commented on the manuscript.

**Acknowledgements**

We thank National Aeronautics and Space Administration (NASA) and United States Geological Survey (USGS) for providing the L8 data used in this study. We thank E. Rignot at Jet Propulsion Laboratory /California Institute of Technology, and J. M. van Wessem, J.T.M. Lenaerts and S. Ligtenberg at Utrecht University for providing their ice velocity products, and data from surface mass balance (SMB) model and firn densification model (FDM), respectively. The Bedmap-2 data is from the British Antarctic Survey. This work was supported by the National Natural Science Foundation of China (Grant Nos. 41431070 and 41590854), CAS/SAFEA International Partnership Program for Creative Research Teams (Grant No. KZZD-EW-TZ-05), Chinese Academy of Sciences, and by the U.S. National Aeronautics and Space Administration (Grant No. NNX10AG31G).

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
