# Peer review of "Antarctic high-resolution ice flow mapping and increased mass loss"

_The Cryosphere, 2017_

## Referee Comment (RC1) · Anonymous Referee #1 · 29 May 2017

This is a review for the manuscript: Antarctic high-resolution ice flow mapping and increased mass loss in Wilkes Land, East Antarctica during 2006–2015.

The paper deals with two ice velocity maps for the years 2014, 2015 based on Landsat-8 data and an assessment of the mass balance change compared to 2006 (based on an earlier ice velocity map). The work in itself is publishable in principle, however, I have three major concerns in the matter.

(1) There is a second manuscript currently under discussion in The Cryosphere. It was submitted a bit later but both discussion papers are dealing with the exact same topic and are using much of the same data sets:

Gardner, A. S., Moholdt, G., Scambos, T., Fahnstock, M., Ligtenberg, S., van den Broeke, M., and Nilsson, J.: Increased West Antarctic ice discharge and East Antarctic stability over the last seven years, The Cryosphere Discuss., doi:10.5194/tc-2017-75, in review, 2017. The group's velocity data are publicly available (http://nsidc.org/data/NSIDC-0710), though not necessarily as presented in the paper.

My primary concern is that the authors of these two manuscripts seem to come to different conclusions in terms of changes in Antarctica. Each group could not have known about the other, but due to the fact that both groups are publishing in 'The Cryosphere', I encourage the editors to initiate a detailed comparison of key elements of the two manuscripts. Maybe the main authors could be invited to provide an open comment to the other paper.

(2) The authors (and Gardner et al. 2017, in review for that matter) use Rignot et al. 2011 as a reference year map for 2006 and 2008, respectively. The MEaSUREs Antarctica ice velocity map (version 1.1 was used, version 2 is now available) is an undated product that aims to provide continent wide coverage for use in ice sheet modeling. The product description states that data acquired over multiple years were included to maximize coverage. While the focus was on acquisitions around IPY, data acquired as early as 1996 were included. I therefore view a quantitative comparison of this map with a product based on data from a single year as problematic. This is true for both manuscripts currently in review in 'The cryosphere'. Both groups alleviate the problem to some degree by utilizing time series data for areas where they are available (i.e. Amundsen sea). In this context, a MEaSUREs product has since become available that provides annual maps from 2005-2016 (http://nsidc.org/data/nsidc-0720). See also the next comment.

(3) Processing ice velocity form Landsat is not new. In addition to references cited in the manuscript, a recent publication presents the Landsat-8–based ice velocity maps and goes on to integrate the data with data from other sensors to present a much more comprehensive, annual time series for Antarctica spanning a time span of 11 years.

Also, Mouginot et al. 2017 is a published manuscript as opposed to a discussion paper. The maps are publicly available at NSIDC: http://nsidc.org/data/nsidc-0720 and are much better suited for the comparative work w.r.t. mass balance.

Mouginot, J., Rignot, E., Scheuchl, B. and Millan, R., 2017. Comprehensive Annual Ice Sheet Velocity Mapping Using Landsat-8, Sentinel-1, and RADARSAT-2 Data. Remote Sensing, 9(4), p.364, doi:10.3390/rs9040364.

Given the close proximity in time of the public availability of these three manuscripts, I consider the continent-wide processing of Landsat-8 data a publication worthy contribution. The novelty of this product is not particularly great, though. The Mass balance assessment is of interest, but the different conclusions of the two papers currently in discussion is confusing. This issue needs to be addressed in some form.

Specific comments:

Line 59: grounded based – > ground based

Line 69-71: While I consider this a contribution worth publishing, please see the two manuscripts mentioned earlier dealing with the same data set.

Line 72: Rignot et al. 2011 is used to estimate the mass discharge in 2006 See my comment above on this topic.

Line 93: "Compared to the satellite interferometric SAR data, the L8 panchromatic imagery is more suitable to estimate ice motion in fast-flowing regions for several reasons,..." While I agree that Landsat-8 is a valuable resource for ice velocity monitoring, I challenge the statement made here based on Mouginot et al. 2017, who show that the error for a single image pair is smaller for SAR when compared to optical.

Line 94, 95: Statement (1) nadir look: Ice velocity from SAR AND optical are generated from data pairs acquired in the same viewing geometry. If the authors mean that velocity maps from spatially adjacent scenes need to be carefully combined because the viewing geometry changes, the statement needs to be clarified. Topographic artifacts

in SAR based ice velocities in Antarctica are regionally limited to mountainous areas like the Antarctic Peninsula and the Transantarctic Mountains. Over large glaciers and ice streams this is less of an issue.

Line96 ff: Statement (2) Non-cloud free sensor. Change 'non-cloud free' to 'cloud cover sensitive' The quality and coverage of the map is owed to a very generous acquisition plan by USGS (i.e. near continuous Landsat-8 acquisitions), so cloud covered images can be discarded and there is still enough data available to provide a near full coverage of the observed area if data from a full year are combined. The fact that more data are available does not support the statement that Landsat-8 is better suited for ice velocity mapping (see my comment above and the assessment in Mouginot et al. 2017)

Line 101 ff: Statement (4) Feature tracking vs speckle tracking vs InSAR phase analysis Speckle Tracking and SAR: Range and azimuth resolution is variable in SAR, but the statement that azimuth is generally lower is not true. In fact, this is mode dependent (see RADARSAT-2 vs Sentinel-1). InSAR phase analysis and SAR: While it is correct the InSAR phase is only sensitive to line of sight displacement, it has been shown that the combination of ascending and descending data leads to a superior result. Combining SAR data from multiple angles even allows a reconstruction of the 3-d flow. Joughin, I. (2002), Ice sheet velocity mapping: A combined interferometric and speckle tracking approach, Ann. Glaciol., 34, 195–201. Gray, L. (2011), Using multiple RADARSAT InSAR pairs to estimate a full three-dimensional solution for glacial ice movement, Geophys. Res. Lett., 38, L05502, doi:10.1029/2010GL046484.

Line 122: See comment for Line 72

Line 156: 100 m resolution product. Does this refer to 100 m posting (i.e. a calculated value every 100 m), instead?

Line 167: These filters factor into the resolution of your product (see comment above).

Line 189, 190: 'In fact, the offset tuning is often called absolute calibration of the ice

velocity data.' Is often called . . . by whom? Please provide reference(s).

Lines 207, 208; Equation 1: I would have expected a weighted average of speeds here. The equation as written will provide more weight to data with longer time separation, but this could be presented more clearly in my opinion (i.e. specify the weights). Also, as written, the authors do not provide an option for spatial adjustment of weights depending on glacier speed for example. Longer time separation will provide advantages in areas of slow flow, but are less suitable in areas of fast flow (see Mouginot et al. 2017)

Equations 2 and 3 are derived from Equation 1. A clearer formulation of the weighting scheme used would necessitate a revision of these equations.

Chapter 4: Decadal glacier dynamics

The primary concern here is that Rignot et al. 2011 is an undated reference map for the period of IPY (definitely not 2006), where data from 1996 were used to provide increased coverage (this aspect is described in Rignot et al. 2011 as well as in the product description at NSIDC). A better comparison here would be using the recently published annual maps provided by the group (see initial comment for access).

Chapter 5: Decadal variations of mass discharge and mass balance

How do the authors account for surface elevation change or ice sheet thinning? e.g. Pritchard, H.D., Arthern, R.J., Vaughan, D.G. and Edwards, L.A., 2009. Extensive dynamic thinning on the margins of the Greenland and Antarctic ice sheets. Nature, 461(7266), pp.971-975.

Do the authors account for basal melting? This is an issue where the grounding line used is outdated and too far downstream.

The authors use multiple sources for grounding lines, but do not appear to have selected a flux gate for estimates well upstream of the GL to improve the flux estimate. This method is described in Mouginot, J., Rignot, E. and Scheuchl, B., 2014. Sustained

increase in ice discharge from the Amundsen Sea Embayment, West Antarctica, from 1973 to 2013. Geophysical Research Letters, 41(5), pp.1576-1584.

Bedmap-2 is a somewhat unreliable source for ice thickness for the purpose of flux measurements due to interpolation issues. The authors seem to access also underlying ground penetrating radar flight lines, however, the choice of gates (i.e. use specific grounding line products) indicates that Bedmap thickness is used. This aspect should be clarified and may require an update of the uncertainties.

The differences to Gardner et al. 2017 (in review) should be investigated further. Both papers deal with the same topic, use roughly the same data sets, but come to different conclusions.

Conclusions: Lines 438,439: The two maps generated do not cover all of Antarctica, the first sentence is therefore misleading.

Figures

Figure 1: Caption should indicate that the InSAR derived velocity is previously published work from someone else. The 2015-2014 difference map shows large differences in several areas where none are expected. This, in my mind, indicates that the overall quality estimate for this data set is likely overly optimistic.

Figure 2: Caption should indicate that the InSAR derived velocity is previously published work from someone else.

Figure 3: Upper inset (this work) shows larger deviations from in situ data than lower inset (Rignot et al. 2011). This runs counter an argument the authors make in lines 93-104, despite the fact that apparently more data pairs were available in 2015 (Landsat-8) compared to 2006 (InSAR). Reference Year (2006) is not a good choice (see comment above).

Figure 4: See chapter 4 comments.

Figure 5: See chapter 5 comments. Also, please compare to Figure 9 from Gardner et al. 2017 (in review) Differences are striking given that essentially the same data were used to generate the results.

Figure 6: Need to cite the sources for data if not generated as part of this work

Figure 7: Given that the velocity difference map presented in Figure 1 shows some massive differences over Ross and Ronne Ice Shelves, I question the accuracy of the data over ice shelves. One way would be to provide a separate quality assessment for ice shelves and provide new error estimates for the region. Absolute differences are interesting, however, I would suggest that the changes are also provided in % of the speed of the glacier/shelf.

Supplementary material: —————————————-

Tables S1 to S4 require a caption. Information sourced from other papers, data sets or sources need to be credited as such.

Line 152 of SM_v16 points to a footnote that does not exist. It seems to be a citation in a wrong format.

Line 225 contains a wrong reference

Table S1, Table S2: How is the velocity measured? Through a single data point, averaged over the gate, or through some other average? More details need to be provided. The naming convention could be described somewhere. Some of the names used could be shown in the figures of the supplementary materials for better orientation.

Table S4: Needs further clarification. There are multiple entries of the same reference showing different values. At the very least provide a comment column. The referenced papers used for this table do not appear in the SOM reference list.

---

## Referee Comment (RC2) · Anonymous Referee #2 · 6 Jun 2017

Review of: Antarctic high-resolution ice flow mapping and increased mass loss in Wilkes Land, East Antarctica during 2006–2015

By Shen et al. for The Cryosphere Discuss. (Ms. Ref. No tc-2017-34)

General Comment

This paper by Shen et al. presents and provides an analysis of two new Antarctic ice sheet wide ice velocity maps derived from an archive of Landsat 8 images acquired in the period 2013-2016. The ice velocity is computed using an optimised automated feature tracking and mosaicking technique. The new velocity maps, together with a previously published InSAR derived velocity map, are combined with ice thickness data from

BEDMAP and modelled surface mass balance (SMB) from RACMO2.3 to calculate ice discharge rates and mass balance for different sectors and individual glaciers. The authors report an increased mass discharge for one sector of East Antarctica (Wilkes Land), which they attribute to widespread accelerating glaciers, while no significant changes are observed in the other 'oceanic' regions. According to the study the same region in East Antarctica also shows the largest change in mass balance, suggesting the risk of irreversible destabilisation of this region.

While the authors did a fine job in creating these new ice velocity maps, which provide a welcome update of previously derived ice velocity maps and which highlight the complex (temporal) patterns of ice flow dynamics in Antarctica, there are several major issues that require careful consideration and major revisions and re-analysis of the data. As already pointed out by reviewer number 1: Gardner et al (ref no tc-2017-75) use pretty much the same source data and methods but come to nearly opposite conclusions: flow acceleration in West Antarctica and 'remarkably' stable glaciers in the East. I am sure the reader/scientific community is eager for an explanation of these contradicting conclusions and I strongly support reviewer's 1 suggestion for both authors to provide an open comment to the other paper.

My foremost concern, however, and also pointed out by reviewer number 1, is the fact that the velocity, discharge and mass balance comparisons are done with a reference dataset that is, frankly speaking, not suitable for the purpose for a number of reasons. The used dataset (Measures; Rignot et al., 2011) is an assembly of ice velocity maps acquired from multiple satellite missions with a temporal range covering more than a decade (see metadata description). While the velocity map is a nice looking and nearly gapless product useful for various purposes and can for instance be used as a rough indication of rapidly changing areas, the large temporal span precludes the use of it for change detection pinpointed to a single year as is done in the study. Additionally, the relative coarse grid spacing, which is not so much of concern for the large ice streams in EAIS and WAIS, is not suitable for comparisons of ice velocity of the many small

glaciers that are found in the (northern) Antarctic Peninsula and that are not well resolved. Slightly different processing settings could lead to very different results when inter-comparing with the newly derived L8 maps. Furthermore, focusing again on the Antarctic Peninsula as the region has one of the largest uncertainties in mass balance, the discharge and mass balance calculations here are also hampered by a lack of suitable ice thickness data. For many of the glaciers BEDMAP ice thickness is too coarse or based on interpolation without actual RES data requiring a careful check for each and every single flux gate with other sources of ice thickness. While it is a straight forward calculation using these datasets, calculating discharge or mass balance (and changes) here requires accurate, detailed and well-defined gates and velocities pinpointed to distinct time periods, otherwise it is a rather meaningless number and not the improvement that is actually needed. I have the major worry that many of the surprising and rather extraordinary results (e.g. tens of glaciers in the Peninsula are reported to have accelerated by more than 600%, fig. 4, line 315) are simply the result of the issues described above. I would recommend focusing on comparing smaller areas for which accurate gate cross sections are available and with ice velocity with well-defined short periods. Below follow some additional comments.

Specific Comments

Ln 78: "the decadal changes can be easily found" – not with this dataset (see above)

Ln 93-104: This paragraph seems a bit biased listing only positive aspects of optical feature tracking versus the limitations of using SAR data. One could just as well argue SAR data is more suitable being an all-weather, year-round technique (important for time series), the possibility of detecting sub-surface features and a capability to derive true 3D velocities. Also, what about slower moving terrain?

Ln 123-131: The accuracy assessment seems rather limited using only GPS (and other data) in slow moving terrain and also from a (in some cases) much earlier period. What does this say about the accuracy in fast flowing terrain at the margins, crucial for

accuracy assessment of the IOM method applied in the study. Where any comparisons on faster flowing ice performed with contemporaneous data sets. What about sensor cross-comparisons, assessment of velocities on ice free terrain etc?

Ln 132-157, section 2.2: Missing any mention of co-registration here. Was this performed/why not?

Ln 156: "100-meter resolution" -> I think grid spacing is meant here.

Ln 162: "or rise" -> What is a rise in this context?

Ln 183-184: "The . . . experiments" unclear what is meant here and actually how the pairs are selected to ensure we still have a distinct "2014" and "2015" map.

Ln 222: "spatial resolution"- not the same as grid spacing

Ln 229-231: "In . . . Antarctica": better move to methods section, it is not a result.

Ln 236: "resolution" - grid size

Ln 245: "conservatively set to be 1/25" - This seems not so conservative. Was this checked e.g. in stable terrain, or just assumed?

Ln 239-264: The uncertainty analysis seems to describe only mis-registration, what about other sources of error?

Ln 270-276: See comment above. Only slow-moving areas are used, could be very different on faster ice streams and glaciers used for the discharge/mass balance analysis.

Ln 280-281: Just looking at the histogram the agreement appears to be better/less skewed for InSAR derived velocity. Referring back to paragraph Ln 93-104 how should we interpret this?

Ln 292-295: A majority of the glaciers accelerated by more than 200% in the northern part of the Western AP: If true, this is a major finding that requires more careful checking. Did you check cross profiles for individual glaciers? Is it not just an artefact from the coarser gridding or different algorithm settings?

Ln 315: See comment above, here it is written that most glaciers in the Western AP actually sped up by more than 600% (nearly all dots are white in this region). This really is an extraordinary result and needs to be checked/discussed in more detail.

Line 322: "for 2006" - > not just 2006, see previous comments.

Line 356-360: "In the Antarctic Peninsula, there was a positive mass balance ($33\pm21$ Gt yr-1) in 2015, contrary to previously studies" -> This statement requires clarification and is in contradiction with other estimates (e.g. from GRACE). Seems impossible considering that so many glaciers have sped up by >600%!

Line 356-360: "probably due to a larger estimate of snow accumulation rate". How much larger? Need to provide numbers to clarify the statement.

Ln 377-379: "likely linked to the incursion of warm CDW" - Any further evidence to support this claim? From figure 7 the PTM in this area doesn't strike me as being particularly high. Is there a relation between glacier thickness, speed up and ocean temperature data here?

Fig. 7: PTM color scale very unclear – seems to have 2 parts with greenish colors.

SM Ln 52: SMB is not the same as surface snow accumulation

SM Ln 69: "total SMB of the Antarctic ice sheet is 1,901 Gt yr-1" – for which year or long-term average – needs clarification.

SM Ln 174: Drygalski Ice Tongue not 'ice shelf'

SM Ln 298: Talev glacier is West coast and not in the Larsen B catchment.

---

## Author Comment (AC1) · 17 Jun 2017

**Reply to the Referee #1's comments**

*Point-by-Point Responses*

1. Q …My primary concern is that the authors of these two manuscripts seem to come to different conclusions in terms of changes in Antarctica. Each group could not have known about the other, but due to the fact that both groups are publishing in 'The Cryosphere', I encourage the editors to initiate a detailed comparison of key elements of the two manuscripts. Maybe the main authors could be invited to provide an open comment to the other paper

R. Here, we carefully compared our manuscript and the manuscript by Gardner et al focusing on conclusion and processing method, and provided the detailed summaries.

*Summary for the conclusions*

(1) Although the two manuscripts seem to have different conclusions, but they actually have the same or similar value estimates. For example, for the total increased ice discharge in entire Antarctic ice sheet, our estimate is 49±40 Gt/yr, close to 35±15Gt/yr from Gardner et al. However, we didn't emphasize the finding that the total ice discharge was increased in entire Antarctic ice sheet during the surveyed period because the uncertainty of our estimate is large compared to the estimate itself. It should be noted that uncertainty of 15Gt/yr may be underestimated by Gardner et al. Please check their Table2, the uncertainty should be 56Gt/yr instead of 15Gt/yr.

(2) For the total mass loss of Antarctic ice sheet, our estimate is 205±90 Gt/yr (an average of 181±68 Gt/yr for 2006 and 230±60 Gt/yr for 2015). The estimate of Gardner et al is 186±93 Gt/yr. The two estimates are quite similar.

(3) We both find that the largest imbalance occurred in Amundsen Sea sector. Our mass balance estimate is –212±24 Gt/yr while Gardner et al's estimate is –213±51 Gt/yr.

(4) Gardner et al. found that the increased ice discharge in Antarctica is mainly contributed by Amundsen Sea Embayment and Getz of west Antarctica, and Marguerite Bay of Antarctic Peninsula. However, we also found the increased ice discharges in these area, for example, 13±1 Gt/yr for Amundsen Sea Embayment (basins 21 and 22), 10±25 Gt yr$^{-1}$ for West Antarctica and 14±7 Gt yr$^{-1}$ for Antarctic Peninsula although the magnitude of increased ice discharges are slightly different to those by Gardner et al. The causes may consist of different grounding lines and method of ice flux used (which will be described in processing method section). Due to large uncertainty of our estimate for West Antarctica, we didn't emphasis the finding that the ice discharges were increased in the area. Furthermore, we also found the rate of accelerated ice discharge in Amundsen Sea Embayment during the surveyed period is less than the former period (1996-2006) (see Figure S4 and S5). As stated

above, the uncertainties for discharge changes may be underestimated by Gardner et al. in their Table2 since the error propagation law was not used for the evaluation of the uncertainties.

***Summary for the processing methods.***

*Method of ice velocity product*

(1) For the ice velocity extraction, we used the COSI-Corr procedure, a frequency-based method developed by Leprince et al., 2007 while Gardner et al. used auto-RIFT technique, a spatial domain one. According to Leprince et al., 2007, the frequency-based technique is generally more accurate than the spatial domain method.

(2) We produced the ice velocity in 100m spatial resolution based on the spatial resolution (~100m) of displacement vectors, while Gardner used the displacement vectors in different spatial resolution (from 240m to 2km) to combine to produce the velocity map of 240*240m grids according to their description in L80.

(3) For quality control for displacement vectors, we used the following three steps to enhance the signal and exclude unreliable measurements. First, an adaptive filter and a median filter were used. They can maximize to remove 'salt and pepper' noise in displacement vectors. The areas covered by cloud and water were excluded from the displacement scenes using integrated QA band. The areas covered by cloud and water also showed lower SNR, generally lower than 0.5, and they were also masked in mosaic product using the SNR <0.9. In addition, the edge of displacement scene was also trimmed. For Gardner et al's manuscript, the quartile filter was used for stable surface, we don't understand whether the method was also applied in fast flow area similarly? Using quartile filter may be not reasonable when the displacements varied with the time separation of image pairs.

  (4) For absolute calibration of ice velocity, we used existing SAR ice velocity as a reference to define stable areas (ice velocity <5m/yr). We considered that the threshold is a relatively rigorous. We assumed that the displacement gradient is stable due to stable image geometry, so the mean values of differences between our ice velocity and InSAR ice velocity were applied for the absolute calibration of mosaic ice velocity product. The calibration processing for NSIDC LISA in Gardner et al was similar to our method, but they used the thresholds <10m/yr and 10-25m/yr respectively to estimate the offsets caused by geolocation error.

(5) In mosaic ice velocity production, we used the time-separation for each image pair as a weight (see equation 1) to estimate ice velocity in order to suppress short-interval velocity measurement because short interval usually has higher error in ice velocity extraction. Generally the coregistration error is independent on the time separation of image pair, in other words, coregistration error of displacement vectors inferred from short time-separation image pair will be amplified largely. NSIDC

LISA were processed in the similar manner in ice velocity extraction but for different weighting (0.3 for 16-day separation, 0.6 for 32-day separation).

*Ice discharge analysis*

(1) We used the Antarctic grounding lines provided by Depoorter et al.(2013) for ice flux (discharge) processing and minor adjustments were made for calculating ice fluxes at different time periods due to incomplete ice thickness and ice flow data as described in section 1 of SI. The grounding line is compilation of near all existed grounding lines products inferred from different techniques (such as optical imagery, InSAR, altimetry) and is considered to be the most approximation of real grounding line. Because the discharge estimate generally varies 3%-8% due to different gate placement from previously studies (depoorter et al. (2013), rignot et al. (2013) and our experiment (a case shown in response to referee #2)), the induced uncertainty is too large for ice discharge estimate. While Gardner used three datasets of grounding lines (namely, GL0, FG1, FG2), and used FG2 to estimate the ice discharge in Antarctic ice sheet while FG2 is far from the true grounding line, especially in West Antarctica. The ice discharge was thus adjusted using the SMB and unmeasured flux due to ice flow convergence/divergence (in L260).

(2) In ice discharge estimate, we considered the direction of ice velocity and plane of flux gate to minimize the influence of inaccurate grounding lines (see figure R1). We considered there may be some errors in defining the grounding lines which may cause that some of ice velocity along grounding line point towards ice sheet instead of the outlet glaciers (case 2). So the derived incorrect flux values cannot be considered and should be removed from the ice flux estimates. Here, for simplification and clarity, we described the two cases (see figure R1) to further explain our processing method, as shown in Figure R1. In case 1, the grounding line is placed on ice-shelf mistakenly, the ice flux through the gate should be zero because the gate and direction of ice flow are parallel. In case 2, the gates b and c are placed by mistake, the ice fluxes in gates b and c can be completely compensated if the effect of ice flow convergence /divergence is negligible. In the next, we further discuss how to calculate the ice flux using the unreal grounding line (Figure R2). In Figure R2, using the real grounding lines, the ice flux of the Fisher, S1, and S2 glaciers should be calculated by the use of the gates of 'L0-4',' 0-1', and '2-3' respectively along the real grounding lines. But in the case of unreal grounding lines used, for the Fisher glacier, the unmeasured ice flux by the gate A (L0-L1) is also measured in gate B ('L1-L2') eventually. For S1, and S2 glaciers, their unmeasured ice flux by the gate A are also measured eventually in Gate B ('L1-L2'). Finally, the ice fluxes from the three glaciers can still be estimated correctly if the ice thickness and velocity data is accurate. The only discrepancy in the estimates for ice flux along different grounding lines is that the additional contribution from the SMB and ice dynamics in the gap area between flux gate and real grounding line. However it is generally small. We believe that our processing method could minimize the effects of inaccurate grounding lines.

[Figure]

Case 1

Case 2

Figure R1 Schemes showing the influence of grounding line error on flux estimate.

[Figure]

Figure R2. A case of ice flux estimate (brown line-real is grounding line, dashed gray line between) L0 and L3 is wrong grounding line)

(3) In Antarctic Peninsula, ice flux was prescribed to have no change during the surveyed period by Gardner et al. while it was estimated based on the InSAR derived velocities (2008) and our L8 based estimates. Since the Landsat ice velocity don't cover the area >82.4 degree in south latitude, Gardner et al used two SAR maps in 1997 and 2009 to extrapolate the ice flux in 2015. We only used one map of 2008 for the ice velocities of 2014 and 2015 under the assumption of no significant change from 2008 to 2015.

(4) We focused on ice discharge and its change on six oceanic sectors although the results were given and the analyses were done on three areas (East, West and Antarctic Peninsula). However, Gardner et al. focused on the latter areas.

2. Q …I therefore view a quantitative comparison of this map with a product based on data from a single year as problematic. This is true for both manuscripts currently in review in 'The cryosphere'. Both groups alleviate the problem to some degree by utilizing time series data for areas where they are available (i.e. Amundsen sea). In this context, a MEaSUREs product has since become available that provides annual maps from 2005-2016 (http://nsidc.org/data/nsidc-0720).See also the next comment

R. We agree your point of view on SAR ice velocity, but it is difficult to discern the exact observation date for SAR ice velocity. We used the reference year for 2006 which are based on the published paper by Rignot et al.(Rignot et al ,NGeo, 2008). In this paper, they analyzed mass balance and its change between 1996 and 2006. Mouginot et al (2012) indicated the spare coverage in 1996 by SAR images. These lead us to select 2006 as reference year. Although the Rignot group just released updated SAR ice velocities on 25, April, 2017, since the released date is later to our submission date, we didn't use the new data. In fact, we used the InSAR velocity data covering 2007, 2008 and 2009 from Rignot et al. (2011), so '2006' should be changed into '2008'. The single year '2008' used is just for convenience, it actually denotes 2007, 2008 and 2009. These would be indicated in the text.

3.Q Processing ice velocity form Landsat is not new. In addition to references cited in the manuscript, a recent publication presents the Landsat-8–based ice velocity maps and goes on to integrate the data with data from other sensors to present a much more comprehensive, annual time series for Antarctica spanning a time span of 11 years. Also, Mouginot et al. 2017 is a published manuscript as opposed to a discussion paper. The maps are publicly available at NSIDC: http://nsidc.org/data/nsidc-0720 and are much better suited for the comparative work w.r.t. mass balance

R. We also noted a new paper by Mouginot et al. (2017) in remote sensing, but the annual ice velocity products can not cover the entire Antarctic ice sheet before 2013.

***Point-by-Point Response for Specific Comments.***

4. Q Line 59: grounded based – > ground based

R. corrected, thanks.

5. Q Line 69-71: While I consider this a contribution worth publishing, please see the two manuscripts mentioned earlier dealing with the same data set.

69 Therefore, here we intend to construct two present-day ice flow maps covering the years of

70 2014 and 2015 for all of the Antarctica inferred from Landsat 8 (L8) images acquired by the
71 Operational Land Imager (OLI).

R. we don't understand the question. We guess that you mean that why the two manuscripts have different conclusions using same data. The question has been replied in response 1.

6. Q. Line 72: Rignot et al. 2011 is used to estimate the mass discharge in 2006 See my comment above on this topic.

R. the reference year for SAR ice velocity has been described in response 2.

7. Q. Line 93: "Compared to the satellite interferometric SAR data, the L8 panchromaticimagery is more suitable to estimate ice motion in fast-flowing regions for several reasons,..." While I agree that Landsat-8 is a valuable resource for ice velocity monitoring, I challenge the statement made here based on Mouginot et al. 2017, who show that the error for a single image pair is smaller for SAR when compared to optical.

R. Agree and thanks. We delete the description of the comparison between Landsat8 and SAR, and the descriptions on characteristics of Landsat 8 move into the former paragraph.

8. Q Line 94, 95: Statement (1) nadir look: Ice velocity from SAR AND optical are generated from data pairs acquired in the same viewing geometry. If the authors mean that velocity maps from spatially adjacent scenes need to be carefully combined because theviewing geometry changes, the statement needs to be clarified. Topographic artifacts in SAR based ice velocities in Antarctica are regionally limited to mountainous areaslike the Antarctic Peninsula and the Transantarctic Mountains. Over large glaciers and ice streams this is less of an issue

R. Agree and thanks. As in response 7, we delete the comparison with SAR.

9. Q. Line96 ff: Statement (2) Non-cloud free sensor. Change 'non-cloud free' to 'cloud cover sensitive' The quality and coverage of the map is owed to a very generous acquisition plan by USGS (i.e. near continuous Landsat-8 acquisitions), so cloud covered images can be discarded and there is still enough data available to provide a near full coverage of the observed area if data from a full year are combined. The fact that more data are available does not support the statement that Landsat-8 is better suited for ice velocity mapping (see my comment above and the assessment in Mouginot et al. 2017)

R. Agree and thanks.

10. Q. Line 101 ff: Statement (4) Feature tracking vs speckle tracking vs InSAR phase analysis Speckle Tracking and SAR: Range and azimuth resolution is variable in SAR, but the statement that azimuth is generally lower is not true. In fact, this is mode dependent (see RADARSAT-2 vs Sentinel-1). InSAR phase analysis and SAR: While it is correct the InSAR phase is only sensitive to line of sight displacement, it has been shown that the combination of ascending and descending data leads to a superior result. Combining SAR data from multiple angles even allows a reconstruction of the 3-d flow. Joughin, I. (2002), Ice sheet velocity mapping: A combined interferometric and speckle tracking approach, Ann. Glaciol., 34, 195–201. Gray, L. (2011), Using multiple RADARSAT InSAR pairs to estimate a full three-dimensional solution for glacial ice movement, Geophys. Res. Lett., 38, L05502, doi:10.1029/2010GL046484.

R. Agree and thanks. As in response 7-9, we deleted the comparison with SAR.

11. Q. Line 122: See comment for Line 72

R. Please see response 2.

12. Q Line 156: 100 m resolution product. Does this refer to 100 m posting (i.e. a calculated

value every 100 m), instead?

R. The 100m resolution of image pairs decides that displacement vector resolution is 100m, then that the resolution of ice velocity product is also 100 m.

13. Q. Line 167: These filters factor into the resolution of your product (see comment above).

R. The local sigma and median filters don't change our spatial resolution of ice velocity products.

14. Q. Line 189, 190: 'In fact, the offset tuning is often called absolute calibration of the ice velocity data.' Is often called : : : by whom? Please provide reference(s)

R. We originally mean to use the term in elonics tuner processing to help us to explain the work. Sorry to mislead you. We change to the general term 'absolute calibration'

15. Q. Lines 207, 208; Equation 1: I would have expected a weighted average of speeds here. The equation as written will provide more weight to data with longer time separation, but this could be presented more clearly in my opinion (i.e. specify the weights). Also, as written, the authors do not provide an option for spatial adjustment of weights depending on glacier speed for example. Longer time separation will provide advantages in areas of slow flow, but are less suitable in areas of fast flow (see Mouginot et al. 2017)

R. Here, we used the time separation as weight to calculate weighted average of ice velocity. The equation 1 is commonly used in high-accuracy InSAR processing. The displacement vectors in a shorter time separation have larger uncertainty in estimate of the ice velocity since the accuracy of co-registration is usually thought to be independent to time interval of two images. Thus, for the longer time interval displacement vectors have lower uncertainty in estimate of the ice velocity, they should have larger weight in ice velocity extraction. The reviewer may worry that the longer time separation will lead to loss of coherence between two images. However, unreliable displacement measurements were excluded after a rigorous quality control as stated in section 2.3 of main text. We also obtained a large numbers of displacement vectors for the shorter time intervals, to assure that the ice velocity can be correctly estimated.

16. Q. Equations 2 and 3 are derived from Equation 1. A clearer formulation of the weighting

Scheme used would necessitate a revision of these equations

R. The equation 1 can be rewritten in the following for better clarity to express time-separation weighted in estimate of ice velocity. Since we gave the errors of displacement vectors, the uncertainty of ice velocity on each grid can be calculated using equation 2 and 3.

$$v = \frac{\triangle t_1 * \dfrac{\triangle d_1}{\triangle t_1} + \triangle t_2 * \dfrac{\triangle d_2}{\triangle t_2} + \cdots + \triangle t_n * \dfrac{\triangle d_n}{\triangle t_n}}{\triangle t_1 + \triangle t_2 + \cdots + \triangle t_n}$$

Figure R3. The rewritten equation 1

17. Q. Chapter 4: Decadal glacier dynamics The primary concern here is that Rignot et al. 2011 is an undated reference map for the period of IPY (definitely not 2006), where data from 1996 were used to provide increased coverage (this aspect is described in Rignot et al. 2011 as well as in the product description at NSIDC). A better comparison here would be using the recently published annual maps provided by the group (see initial comment for access).

R. Although the older SAR ice flow map is a synthesis compilation inferred from SAR images acquired in longer time period, including SAR data in 1996 which has been described in SI, we checked the annual maps newly released by the same group, it is still difficult to obtain full coverage using the annual ice flow map in any single year before 2013. As Rignot et al. (2011) mentioned, the ice velocities first covering entire Antarctica were inferred from the data acquired mainly from 2007 to 2009. Therefore, we agree now to use 2008 as a reference year when SAR ice velocity was compared with our 2015 L8 ice velocity to investigate the ice discharge, mass balance and the changes from 2008 to 2015.

18. Q. Chapter 5: Decadal variations of mass discharge and mass balance, How do the authors account for surface elevation change or ice sheet thinning? e.g. Pritchard, H.D., Arthern, R.J., Vaughan, D.G. and Edwards, L.A., 2009. Extensive dynamic thinning on the margins of the Greenland and Antarctic ice sheets. Nature, 461(7266), pp.971-975.

R. Generally, there are three independent methods to estimate net mass balance of Antarctic ice sheet on satellite techniques. The first one is surface elevation change by satellite radar/laser altimetry, the second is input and output method (IOM) as the manuscript mentioned, and the third is mass change from satellite gravity. Because we estimated the ice discharge along the grounding lines, not FG2 as in Gardner's manuscript, we don't need to take into account the contribution of the surface elevation change in estimate of ice discharge. However, in Gardner's manuscript, contribution from the surface elevation change in the areas between FG2 and GL0 in estimate of ice discharge must be considered. In principle, we agree that dynamic surface elevation change on grounding lines affect ice discharge estimate to some degree, but the surface elevation change is far less than uncertainty of ice thickness, and is thus not considered in our study.

19. Q. Do the authors account for basal melting? This is an issue where the grounding line used is outdated and too far downstream.The authors use multiple sources for grounding lines, but do not appear to have selected a flux gate for estimates well upstream of the GL to improve the flux estimate. This method is described in Mouginot, J., Rignot, E. and Scheuchl, B., 2014. Sustained increase in ice discharge from the Amundsen Sea Embayment, West Antarctica, from 1973 to 2013. Geophysical Research Letters, 41(5), pp.1576-1584.

R. We didn't consider the basal melting. Actually, the basal melt should be considered in estimate of net mass balance. However, due to difficulty for the estimate of change of basal melt, it is still difficult to be involved in the analysis of mass balance change on Antarctic ice sheet. We used the grounding lines provided Depoorter et al. (2013), although it has been updated in Amundsen Sea and Totten Glacier areas since then. We followed the Depoorter et al (2013) and Rignot et al. (2008, 2013) to estimate ice flux along grounding lines and didn't use the method in Mouginot et al. (2014) and Gardner et al. (in discussion, TC), the reasons have been given in Response 2.

20. Q. Bedmap-2 is a somewhat unreliable source for ice thickness for the purpose of flux measurements due to interpolation issues. The authors seem to access also underlying ground penetrating radar flight lines, however, the choice of gates (i.e. use specific grounding line products) indicates that Bedmap thickness is used. This aspect should be clarified and may require an update of the uncertainties.

R. For minimize the uncertainty of Bedmap-2 ice thickness, we also used the IPR data as much as possible as described in section 1 in SI. In uncertainty assessment, we used the error of Bedmap-2 ice thickness to estimate the uncertainty of ice discharge no matter Bedmap-2 data or IPR data were used, so the uncertainty estimate was relatively conservative.

21. Q. The differences to Gardner et al. 2017 (in review) should be investigated further. Both papers deal with the same topic, use roughly the same data sets, but come to different conclusions

R. please see in Response 1.

22. Q. Conclusions: Lines 438,439: The two maps generated do not cover all of Antarctica, the first sentence is therefore misleading.

R. Agree, we change the sentence to 'we constructed two continent-wide high-resolution ice flow maps covering the years of 2014 and 2015 in Antarctica, which…'

***Point-by-Point Responses for Figures***

23. Q. Figure 1: Caption should indicate that the InSAR derived velocity is previously published work from someone else. The 2015-2014 difference map shows large differences in several areas where

none are expected. This, in my mind, indicates that the overall quality estimate for this data set is likely overly optimistic.

R. Thanks, we added the reference for InSAR derived ice velocity. The relatively large changes up to 20 m/yr (such as in Ross and Ronne ice shelves, and some places of the ice sheet) are attributable to the less displacement scenes in 2014 or 2015 ice velocity mosaics, the uncertainty of absolute calibration in ice shelves, especially in fast-flow ice shelves, or interannual variation of ice flow. But these didn't affect ice discharge estimates along grounding lines. The accuracy of Landsat8 ice velocity should be investigated using independent field data (see section 3.3), the gridded uncertainty of ice velocity is based on a given precision of image coregistration, because it is difficult to estimate the uncertainty of ice velocity on each grid.

24.Q. Figure 2: Caption should indicate that the InSAR derived velocity is previously published work from someone else

R. The reference for InSAR derived velocity is added, thanks.

25.Q. Figure 3: Upper inset (this work) shows larger deviations from in situ data than lower inset (Rignot et al. 2011). This runs counter an argument the authors make in lines 93-104, despite the fact that apparently more data pairs were available in 2015 (Landsat-8) compared to 2006 (InSAR). Reference Year (2006) is not a good choice (see comment above).

R. we changed the description in lines 93-104. The reference year has changed to 2008, and see more details in Response 2.

26. Q. Figure 5: See chapter 5 comments. Also, please compare to Figure 9 from Gardner et al. 2017 (in review) Differences are striking given that essentially the same data were used to generate the results

R. Our figure is plotted in term of ice shelf as an evaluated unit while Gardner's is based on basins. Our results are more consistent with the L125 ice velocity in 125m spatial resolution from Gardner et al. 2017 although there remain some discrepancies in some basins (see Figure 6).

27. Q. Figure 6: Need to cite the sources for data if not generated as part of this work

R. we add the references for the bathymetry data and sub-glacial basins used.

28. Q. figure 7: Given that the velocity difference map presented in Figure 1 shows some assive differences over Ross and Ronne Ice Shelves, I question the accuracy of the data over ice shelves. One way would be to provide a separate quality assessment for ice shelves and provide new error estimates for the region. Absolute differences are interesting, however, I would suggest that the changes are also provided in % of the speed of the glacier/shelf.

R. According your comments, we also compared ice velocities with Velmap, InSAR estimates. Here, some cases in some typical ice shelves (Ross, Ronne, Amery, Mertz ice shelves) are shown (Figure R4, R5), the row in Figure R4, R5 shows the histograms of InSAR, L8 2015 and bird view of areas. The histogram is estimated based on the differences between our ice velocities (or InSAR) and Velmap velocity data. From the cases, we can find that the two products don't shown significant different in accuracy. But it is noted that a long term change of ice velocity may be subject to occur due to the long time interval between these ice velocity estimates. We have changed to % of speed according your advices. Thanks.

[Figure]

Figure R4 some cases for the investigation of ice velocities (STD: standard deviation).

[Figure]

Figure R5. Some cases for the investigation of ice velocity.

***Point-by-Point Response for Supplementary material:***

29. Q. Tables S1 to S4 require a caption. Information sourced from other papers, data sets or sources need to be credited as such.

R. the captions of tables S1 to S4 have been added in SI, and data sources from other papers have been cited and checked.

30. Q. Line 152 of SM_v16 points to a footnote that does not exist. It seems to be a citation in a wrong format

R. We apologize for our negligence. Corrected. Thanks.

31. Q. Line 225 contains a wrong reference

R. Apologize again, corrected.

32. Q. Table S1, Table S2: How is the velocity measured? Through a single data point, averaged over the gate, or through some other average? More details need to be provided. The naming convention could be described somewhere. Some of the names used could be shown in the figures of the supplementary materials for better orientation.

R. The velocity are picked on the crossover between grounding lines and a profile along ice flow central line, which is placed from the upstream of glacier to outlet of ice shelf or glacier. Generally, the sampled velocity value is averaged ice velocity in 3 pixels (about 300m) around the crossover along the grounding line. According to your comments, some naming convention used in main text or SI now keep consistent as in Tables to avoid confusing. For example, 'mass' (2015) is changed to 'GLF' (2015) in Table S2.

33. Q. Table S4: Needs further clarification. There are multiple entries of the same reference showing different values. At the very least provide a comment column. The referenced papers used for this table do not appear in the SOM reference list.

R. Sorry for confusing, there are results in different time period or different observation techniques in a same paper. We have carefully checked the references, the majority of results were cited form Shepherd et al. (2012) and the rests are cited in Main text. Thanks.

---

## Author Comment (AC2) · 17 Jun 2017

**Reply to the Referee #2's comments**

*Point-by-Point Responses*

1. Q. …As already pointed out by reviewer number 1: Gardner et al (ref no tc-2017-75) use pretty much the same source data and methods but come to nearly opposite conclusions: flow acceleration in West Antarctica and 'remarkably' stable glaciers in the East. I am sure the reader/scientific community is eager for an explanation of these contradicting conclusions and I strongly support reviewer's 1 suggestion for both authors to provide an open comment to the other paper.

R. This comment was just proposed by reviewer 1, and we have carefully answered the question. Please check our response 1 to Referee #1. Thanks.

2. Q. My foremost concern, however, and also pointed out by reviewer number 1, is the fact that the velocity, discharge and mass balance comparisons are done with a reference dataset that is, frankly speaking, not suitable for the purpose for a number of reasons. The used dataset (Measures; Rignot et al., 2011) is an assembly of ice velocity maps acquired from multiple satellite missions with a temporal range covering more than a decade (see metadata description). While the velocity map is a nice looking and nearly gapless product useful for various purposes and can for instance be used as a rough indication of rapidly changing areas, the large temporal span precludes the use of it for change detection pinpointed to a single year as is done in the study.

R. We agree your point of view on SAR ice velocity, but it is difficult to discern the exact observation date for SAR ice velocity. We used the reference year for 2006 which are based on the published paper by Rignot et al.(Rignot et al ,NGeo, 2008). In this paper, they analyzed mass balance and its change between 1996 and 2006. Mouginot et al (2012) indicated the spare coverage in 1996 by SAR images. These lead us to select 2006 as reference year. Although the Rignot group just released updated SAR ice velocities on 25, April, 2017, since the released date is later to our submission date, we didn't use the new data. In fact, we used the InSAR velocity data covering 2007, 2008 and 2009 from Rignot et al. (2011), so '2006' should be changed into '2008'. The single year '2008' used is just for convenience, it actually denotes 2007, 2008 and 2009. These would be indicated in the text.

3. Q. Additionally, the relative coarse grid spacing, which is not so much of concern for the large ice streams in EAIS and WAIS, is not suitable for comparisons of ice velocity of the many small glaciers that are found in the (northern) Antarctic Peninsula and that are not well resolved. Slightly different processing settings could lead to very different results when inter-comparing with the newly derived L8 maps.

R. We found the change of ice velocity in northwest Antarctic Peninsula is very large due to low estimates of SAR velocities in 2008 while the low estimates of SAR velocities may be caused by relative coarse grid in SAR data as you mentioned. Gardner et al thought that they are caused by interpolation. Because the narrow outlet glaciers can still be found in that product of Rignot et al. (2011), interpolation may be not an only cause. A detailed comparison between our velocity and SAR

velocity can be found in Response 20. Taking into account the large uncertainties of SAR velocity data, SMB and thickness data, although we estimated the ice discharges, mass balances and the changes in the northwest Antarctic Peninsula we did not draw conclusions there.

4. Q. Furthermore, focusing again on the Antarctic Peninsula as the region has one of the largest uncertainties in mass balance, the discharge and mass balance calculations here are also hampered by a lack of suitable ice thickness data.

R. As you said, the Antarctic Peninsula has large uncertainty caused by multiple factors, such as SMB, ice thickness and ice velocity (spatial resolution), surface runoff, and basal melting, etc. So we didn't draw a definitive conclusions for the Antarctic Peninsula. In the manuscript, we showed a positive mass balance mainly due to the new high-resolution SMB, we also analyzed the potential causes in main text.

5. Q. For many of the glaciers BEDMAP ice thickness is too coarse or based on interpolation without actual RES data requiring a careful check for each and every single flux gate with other sources of ice thickness. While it is a straight forward calculation using these datasets, calculating discharge or mass balance (and changes) here requires accurate, detailed and well-defined gates and velocities inpointed to distinct time periods, otherwise it is a rather meaningless number and not the improvement that is actually needed. I have the major worry that many of the surprising and rather extraordinary results (e.g. tens of glaciers in the Peninsula are reported to have accelerated by more than 600%, fig. 4, line 315) are simply the result of the issues described above

R. We agree that the BEDMAP ice thickness is relatively coarse, in order to eliminate or suppress its effects, we also used the ERS data, but the ERS data only occupy ~19% of all grounding lines. If as Gardner did in TC Discussion, the grounding lines are moved intentionally toward inland for much more coverage of ERS, the ice discharge maybe influenced. Based on our experiment in Bryd glaciers, the ice discharge could changes as much as 10-20% (~2-4Gt/yr) even if 5-14km inland movement (see Figure R1, GL0, FL1~FL6) and the ERS data are used.

Although SMB and elevation change in the gap area between flux gate and grounding lines can be used to correct the ice discharge from inland flux gate, the total contribution from elevation change and SMB is found to be only ~0.04Gt/yr for the gap between flux gate and grounding line, which is largely less than the magnitude of ice discharge variation due to gate placement movement. So the method moving the gate is skeptical. In addition, theoretically, the elevation change is not completely attributed to dynamic volume change, it is also caused by multiple factors such as firn densification, SMB variation and snow drift by wind, and etc. New error sources are thus also involved, such as for SMB and firn densification. The large acceleration in northwest Antarctic Peninsula is discussed in Response 3.

[Figure]

Figure R1. Grounding line and flux gate placements in Byrd Glacier. (GL0: grounding lines, FL (1-6): the locations of flux gate for sensitivity experiments. The values of ice discharge are shown in braces).

***Point-by-Point Responses Specific Comments***

6.Q. Ln 78: "the decadal changes can be easily found" – not with this dataset (see above)

R. the sentence is changed to "the changes of mass balances in the period can be easily found". The InSAR-derived ice velocity has been mentioned in former sentences.

7. Q. Ln 93-104: This paragraph seems a bit biased listing only positive aspects of optical feature tracking versus the limitations of using SAR data. One could just as well argue SAR data is more suitable being an all-weather, year-round technique (important for

time series), the possibility of detecting sub-surface features and a capability to derive true 3D velocities. Also, what about slower moving terrain?

R. As responded to Referee #1, we agree your point of view. We delete the description of the comparison between Landsat8 and SAR. And the descriptions on characteristics of Landsat 8 are now involved into the former paragraph.

8. Q. Ln 123-131: The accuracy assessment seems rather limited using only GPS (and other data) in slow moving terrain and also from a (in some cases) much earlier period. What does this say about the accuracy in fast flowing terrain at the margins, crucial for accuracy assessment of the IOM method applied in the study? Where any comparisons on faster flowing ice performed with contemporaneous data sets. What about sensor cross-comparisons, assessment of velocities on ice free terrain etc?

R. We originally used all field-surveying measurements in Velmap project to investigate the reliability of our results because it is very difficult find the contemporaneous data sets and the fast flow glaciers may have the velocity change in the time interval. In order to response your concerns, all data compiled in velmap project (hereafter using velmap velocity) and detailed comparison will be given in supplement file. Here, we show some cases. The first one (see Figure R2), in Ronne Ice Shelf close to the vicinity of Evans glacier, in the Label aaa/bbb, the 'aaa' strands for differences between InSAR-derived and Velmap velocities and 'bbb' is the differences between Landsat 8 2015 and velmap velocities. No matter for the high shear regions (blue ellipse), or other regions, the results for aaa and bbb are in general close each other, the high shearing area probably shown continuous slowdown of ice velocity since 2008. This showed the agreement between our L8 velocity and the Velmap velocity is similar to the case for InSAR velocity. This is also found in the north of Larsen B (Figure R3) and the downstream of Bindschadler glacier (Figure R4)

[Figure]

Figure R2. Ronne ice shelf in the vicinity of Evans glacier

[Figure]

Figure R3. The north of Larsen B

[Figure]

Figure R4. The downstream of Bindschadler glacier (in Ross ice shelf).

9. Q. Ln 132-157, section 2.2: Missing any mention of co-registration here. Was this performed/why not?

R. Because we used the released COSI-Corr software to obtain the displacement vector. We didn't discuss the details of the method. The procedure is based on the co-registration in frequency domain. The characteristics of the method are only shown in the SI.

10. Q. Ln 156: "100-meter resolution" -> I think grid spacing is meant here

R. we get displacement vector with 100-meter resolution since the displacement vector is expressed in image model of 100m resolution. Therefore, our ice velocity mosaic has a resolution of the 100-meter.

11. Q. Ln 162: "or rise" -> What is a rise in this context?

R. We mean the rise is similar to island but with smaller size.

12. Q. Ln 183-184: "The : : : experiments" unclear what is meant here and actually how the pairs are selected to ensure we still have a distinct "2014" and "2015" map

R. At first, we did experiments for some glacier areas, we found that the co-registration would fail if time separation is larger than two seasons (for example 2013-2014 summer, and 2014-2015 summer). The '2014' and '2015' are defined in caption of Figure 1, showing the start and end date. To check one couple of image pairs if they fall into a specific year, the time interval must lie between the start and end date of the year. For example, if the displacement vectors are inferred from the start image on

December, 2013, and end image on December, 2014, the result for ice velocity should be incorporated into '2014'.

13. Q. Ln 222: "spatial resolution"- not the same as grid spacing

R. Please see Response 10.

14. Q. Ln 229-231: "In : : : Antarctica": better move to methods section, it is not a result.

R. Thanks, the sentence has been moved to method section.

15. Q. Ln 236: "resolution" - grid size

R. yes, corrected

16. Q. Ln 245: "conservatively set to be 1/25" - This seems not so conservative. Was this checked e.g. in stable terrain, or just assumed?

R. The value is larger than the suggested value (1/50) by Leprince et al. (2007) and 1/128 pixel for SAR velocity by Rignot (2011). Because it is very difficult to get the errors for each grid (pixel), here, we assumed there are same co-registration errors for each displacement scenes. The assumption is reasonable after the check in stable terrain (see section 3.3 in Main text).

17 Q. Ln 239-264: The uncertainty analysis seems to describe only mis-registration, what about other sources of error?

R. The other sources of errors included those for ice thickness, grounding line fluctuation, etc. and are described in SI.

18. Q. Ln 270-276: See comment above. Only slow-moving areas are used, could be very different on faster ice streams and glaciers used for the discharge/mass balance analysis.

R. Please see Response 8.

19. Q. Ln 280-281: Just looking at the histogram the agreement appears to be better/less skewed for InSAR derived velocity. Referring back to paragraph Ln 93-104 how should we interpret this?

R. The difference between our observation and InSAR-derived velocity is probably due to the real surface change in the long period, or the uncertainties of our observation. Same as the Response 7, we delete the description of the comparison between Landsat8 and SAR. And the descriptions on characteristics of Landsat 8 are involved into the former paragraph.

20. Q. Ln 292-295: A majority of the glaciers accelerated by more than 200% in the northern part of the Western AP: If true, this is a major finding that requires more careful checking. Did you check cross profiles for individual glaciers? Is it not just an artefact from the coarser gridding or different algorithm settings?

R. As the response 3, we compared the individual glaciers. InSAR-derived ice velocity is very small, especially in the grounding lines. In the next, we present a case in a glacier in WAP, the outlet glacier generally move fast in the front of the glacier and show in high-shearing area (See Figure R5). In InSAR-derived ice velocity, the velocity in the vicinity of grounding lines is unrealistically low (see Figure R6) while our observation shows more realistic.

[Figure]

Figure R5. A glacier in WAP (white solid line: grounding lines; red lines: the area of the glacier; green line: profile (from upstream to outlet). The red box shows the crossover between grounding line and profiles.

[Figure]

[Figure]

Figure R6 a) Landsat8 2015 ice velocity in the glacier, b) InSAR-derived ice velocity; c) the profiles of ice velocity from upstream to outlet (location: green line in Figure R5, and Figure 6a,6b). The black triangle shows the location of grounding line.

21. Q. Ln 315: See comment above, here it is written that most glaciers in the Western AP actually sped up by more than 600% (nearly all dots are white in this region). This really is an extraordinary result and needs to be checked/discussed in more detail.

R. Please see Response 20 for comparison. Since there are no field measurements available in the area, it is difficult to compare with field data.

22. Q. Line 322: "for 2006" - > not just 2006, see previous comments.

R. please see Response 2.

23. Q. Line 356-360: "In the Antarctic Peninsula, there was a positive mass balance (33±21 Gt yr-1) in 2015, contrary to previously studies" -> This statement requires clarification

and is in contradiction with other estimates (e.g. from GRACE). Seems impossible considering that so many glaciers have sped up by >600%!

R. Because the estimate of SMB used is larger than the previous estimate from low-resolution SMB product (27km or 52 km). It caused the positive mass balance in AP in our estimate. The previous SMB in AP is generally less than 100 Gt/yr, such as 94Gt/yr for Rignot et al. (2008) in Ngeo.

24. Q. Line 356-360: "probably due to a larger estimate of snow accumulation rate". How much larger? Need to provide numbers to clarify the statement

R. please see Response 23.

25. Q. Ln 377-379: "likely linked to the incursion of warm CDW" - Any further evidence to support this claim? From figure 7 the PTM in this area doesn't strike me as being particularly high. Is there a relation between glacier thickness, speed up and ocean temperature data here?

R. For plotting, we used a PTM data beneath 200m and the data was obtained at a long time ago. This may not represent the current state of ocean, especially in EIS sector, but PTM data can show a warm oceanic environment in WAP. However, the recent measurement on Totten glacier may support our claim. Australian's scientists reported warm water reaches the glacier and may be driving melt of the glacier from below. The new observation has been referenced in our manuscript. The more information for the link.
http://www.abc.net.au/news/2015-01-26/sea-water-melting-totten-glacier-in-antarctica-from-below/6047076

26. Q. Fig. 7: PTM color scale very unclear – seems to have 2 parts with greenish colors.

R. We tried many color schemes for high contrast between grounded ice and ocean. Although the bounder between ocean and grounded ice is not easily to discern because the two color scales have same color component, fortunately, the majority of boundary between grounded ice and ocean are filled by cold color scheme (blue and dark blue), which is helpful for the plot's clarity.

27. Q. SM Ln 52: SMB is not the same as surface snow accumulation

R. corrected. Thanks.

28. Q. SM Ln 69: "total SMB of the Antarctic ice sheet is 1,901 Gt yr-1" – for which year or long-term average – needs clarification

R. we used the long-term average value of SMB.

29. Q. SM Ln 174: Drygalski Ice Tongue not 'ice shelf

R. corrected, Thanks.

30. Q. SM Ln 298: Talev glacier is West coast and not in the Larsen B catchment.

R. corrected to Crane glacier, Thanks.

---

## Editor Comment (EC1) · E. Berthier (Editor) · 12 Jul 2017

Dear Authors,

Although both reviewers seemed to agree on the importance of your continent-wide discharge and mass balance assessment, they raised substantial methodological issues. I have been reading carefully your answers to their comments. I appreciate your efforts to reconcile your values with the other similar study in discussion for TC. Unfortunately, I do not think you provided sufficiently convincing responses to warrant consideration of a revised version of your manuscript.

[Figure]

Interactive
comment

Among the main weaknesses of the study are:

(1) The lack of clear time stamp for the velocity map. The availability of a new velocity product with well-defined time stamps should now be taken into account, at least by quantifying the errors involved when attributing all velocity measurements to a single year (2008, and not 2006). I agree that it is unfortunate that this product was available after your submission date, but still it should now be used to improve your study and aim at the best discharge estimate for the ~IPY period.

(2) One major issue is the use of mainly BEDMAP2 as ice thickness data at the grounding line for ice discharge calculation. As reviewer#2 put it "While it is a straight forward calculation using these datasets, calculating discharge or mass balance (and changes) here requires accurate, detailed and well-defined gates and velocities inpointed to distinct time periods, otherwise it is a rather meaningless number and not the improvement that is actually needed." The use of BEDMAP2 can lead to large systematic errors, as recently demonstrated for the Getz and Abbot sectors in Chuter et al. [2017]. It seems also mandatory to take into account the elevation change (mostly thinning) at / close to the grounding line that took place between 2008 and 2015. In term of mass flux, the thinning will possibly partly counteract the effect of the velocity increase. The fact that the total thinning between 2008 and 2014/15 is within uncertainties of the total ice thickness from Bedmap2 is not a good reason to neglect it because it could lead to systematic errors in your discharge assessment. For example, I do not think you neglected the velocity change when they are within error bounds of the velocity measurement.

(3) The highly unrealistic velocity variations in the Antarctic Peninsula (AP). Overall, the total mass balance of the AP (positive in your study) is in very strong disagreement with published results for this area using various techniques. The argument that you "did not draw conclusion" (your reply to reviewer#2) is not a satisfying one. A much more critical discussion is required; otherwise it weakens the rest of your conclusions.

To this list (a synthesis of the major reviewer's comments), I would add the need to
well-justify the use of a long term average (1979–2014 if I understood correctly) of the SMB to measure the total mass balance for two snapshots (2008 and 2014/2015). What justifies ignoring the inter-annual SMB variability?

Further, your main conclusion of increased mass loss in the Wilkes Land would need to be compared thoroughly to other assessments in this sector using different techniques (altimetry, gravimetry, other I/O estimates if available). The need for such a comparison is in fact true for all your study regions to put/back up your findings based on the existing literature. Providing all numbers from the literature in an excel spreadsheet (Table S4) is certainly useful for some readers but does not help to see the agreement/disagreement between yours and previous studies.

Your continent-wide velocity maps and revised ice discharge and mass balance estimates will deserve publication in the future but require some additional data processing that goes beyond the scope of what can be done in the framework of the present submission.

I am sorry for not being more positive. I hope that the reviewer's comments will help you to re-submit a deeply revised version elsewhere.

Please do not hesitate to contact me in case you have any questions.

Best regards, Etienne Berthier – TC Editor

Reference: Chuter, S. J., Martín-Español, A., Wouters, B. and Bamber, J. L.: Mass Balance Reassessment of Glaciers Draining into the Abbot and Getz Ice Shelves of West Antarctica, Geophys. Res. Lett., doi:10.1002/2017GL073087, 2017.
* * *